# In Situ Calibration and Trajectory Enhancement of UAV and Backpack LiDAR Systems for Fine-Resolution Forest Inventory

Tian Zhou [1], Radhika Ravi [1], Yi-Chun Lin [1], Raja Manish [1], Songlin Fei [2] and Ayman Habib [1,*]

1 Lyles School of Civil Engineering, Purdue University, West Lafayette, IN 47907, USA; zhou732@purdue.edu (T.Z.); rradhika@umich.edu (R.R.); lin934@purdue.edu (Y.-C.L.); rmanish@purdue.edu (R.M.)
2 Forestry and Natural Resources, Purdue University, West Lafayette, IN 47907, USA; sfei@purdue.edu
* Correspondence: ahabib@purdue.edu; Tel.: +1-(765)-496-0173

**Abstract:** Forest inventory has been relying on labor-intensive manual measurements. Using remote sensing modalities for forest inventory has gained increasing attention in the last few decades. However, tools for deriving accurate tree-level metrics are limited. This paper investigates the feasibility of using LiDAR units onboard uncrewed aerial vehicle (UAV) and Backpack mobile mapping systems (MMSs) equipped with an integrated Global Navigation Satellite System/Inertial Navigation System (GNSS/INS) to provide high-quality point clouds for accurate, fine-resolution forest inventory. To improve the quality of the acquired point clouds, a system-driven strategy for mounting parameters estimation and trajectory enhancement using terrain patches and tree trunks is proposed. By minimizing observed discrepancies among conjugate features captured at different timestamps from multiple tracks by single/multiple systems, while considering the absolute and relative positional/rotational information provided by the GNSS/INS trajectory, system calibration parameters and trajectory information can be refined. Furthermore, some forest inventory metrics, such as tree trunk radius and orientation, are derived in the process. To evaluate the performance of the proposed strategy, three UAV and two Backpack datasets covering young and mature plantations were used in this study. Through sequential system calibration and trajectory enhancement, the spatial accuracy of the UAV point clouds improved from 20 cm to 5 cm. For the Backpack datasets, when the initial trajectory was of reasonable quality, conducting trajectory enhancement significantly improved the relative alignment of the point cloud from 30 cm to 3 cm, and an absolute accuracy at the 10 cm level can be achieved. For a lower-quality trajectory, the initial 1 m misalignment of the Backpack point cloud was reduced to 6 cm through trajectory enhancement. However, to derive products with accurate absolute accuracy, UAV point cloud is required as a reference in the trajectory enhancement process of the Backpack dataset.

**Keywords:** LiDAR; feature extraction/matching; Backpack MMS; UAV MMS; system calibration; GNSS/INS trajectory enhancement; forest inventory

## 1. Introduction

Forests provide critical ecosystem services (e.g., fiber and timber), but are constantly challenged by various environmental stressors. Data-driven policies and management practices, powered by accurate inventory, are essential for the long-term sustainability of forest ecosystems. Traditionally, forest inventory has been conducted manually, which is expensive and time-consuming. With recent advances in sensor and algorithmic technologies, remote/proximal sensing, including (a) LiDAR and photogrammetry from manned/uncrewed aerial vehicles, (b) stationary terrestrial laser scanners (TLS), and (c) mobile ground LiDAR, has recently been explored as an alternative for automated tree-level inventory at various scales. These sensors/platforms have trade-offs in terms of cost, field survey efficiency, spatial coverage, spatial resolution, and level of detail of the acquired information [1–3].

The forestry research community has shown a keen interest in digital aerial photogrammetry using imagery obtained by manned aerial systems to derive inventory biometrics—e.g., tree height, stem volume, and basal area [4–6]. However, the major limitation of this approach is that image-based point clouds primarily represent the outer envelope of a forest canopy. On the other hand, LiDAR energy can penetrate through gaps between leaves, allowing it to capture data from below-canopy structures. In this regard, manned airborne LiDAR, which provides data with large spatial coverage and fine resolution, is widely used to derive leaf area index (LAI), canopy height, crown dimension, as well as the attributes beneath the canopy, such as stem map and terrain information [7–10].

Compared with manned aerial systems, uncrewed aerial vehicles (UAVs) possess distinct advantages due to their affordability, close sensor-to-object distance, and ease of deployment and operation. These factors enable UAVs to provide high spatial and temporal data effectively. Several studies derived forest biometrics using orthophotos and point clouds generated from UAV images [11–15]. UAV LiDAR matches most advantages of manned airborne LiDAR, except for reduced spatial coverage. Numerous research investigations have applied UAV LiDAR data to segment individual trees and estimate various forest attributes, including canopy cover, tree height, diameter at breast height (DBH), and above-ground biomass [16–23]. However, when conducting UAV LiDAR flights above the forest canopy, the capability to map below-canopy structures deteriorates due to factors such as tree density and leaf cover. Achieving detailed mapping of below-canopy structures, which is crucial for obtaining precise estimates of forest biometrics such as DBH and debris, cannot always be guaranteed.

Ground systems, including TLS and mobile ground LiDAR, offer the ability to acquire detailed information beneath the forest canopy. Previous research utilized TLS data with high spatial resolution to estimate forest structural metrics for individual trees [2,24,25]. For TLS, conducting large-scale field surveys is time-consuming. Moreover, the post-processing of acquired data is complex. On the other hand, mobile ground systems can maneuver within the area of interest to cover large areas [26]. Several studies used ground systems to generate stem maps, derive DBH estimates, and conduct crown segmentation [27–31]. Nevertheless, ground systems are susceptible to occlusions caused by terrain and above-ground features, and obstacles on the forest floor can restrict the movement of the platforms. The main challenge for under-canopy mobile LiDAR surveys is the intermittent access to Global Navigation Satellite System (GNSS) signals. Having continuous access to GNSS signals is critical to deriving reliable trajectory information and, consequently, mapping products with high georeferencing accuracy.

Several studies tackled the mapping in GNSS-denied/challenging environments to derive high-quality LiDAR point clouds for forest inventory. Kukko et al. [32] proposed a trajectory enhancement strategy for a LiDAR system mounted on an all-terrain vehicle. LiDAR points from a short time period were reconstructed using the initial trajectory generated through the processing of data acquired by GNSS/Inertial Navigation System (INS). Then, centroids of tree trunks were extracted from each section of LiDAR data and matched. Corrections to the initial trajectory were estimated by considering two constraints: maintaining the relative position/orientation transformation between two successive epochs in the initial trajectory and minimizing the discrepancies among conjugate tree trunk centroids derived from different LiDAR sections. Although high relative accuracy was achieved for the tree trunk locations, this trajectory enhancement approach has several limitations: (a) it is based on a data-driven approach, while assuming that system calibration parameters and short-term trajectory used to generate an individual LiDAR section are errorless, which is not often the case; (b) since only the relative position/orientation information of the initial trajectory is used, an additional transformation step is required to align the corrected trajectory with the initial one to ensure some level of absolute accuracy, which could lead to aggregated errors; and (c) using only centroids of tree trunks for trajectory enhancement leads to weak control in the vertical direction. Chiella et al. [33] merged information from GNSS, Attitude and Heading Reference Systems (AHRS), and 2D LiDAR odometry for

mobile mapping system (MMS) navigation. The LiDAR-based odometry utilized tree trunk features to derive the motion of the platform through a scan matching algorithm. Their work's limitations include: (a) it only evaluated the positional trajectory parameters, while the orientation parameters were not estimated, and (b) it cannot deal with 3D LiDAR, since the scanning mechanism of 2D LiDAR formed the basis for their odometry strategy.

Tang et al. [34] investigated a Simultaneous Localization and Mapping (SLAM)-aided positioning solution using point clouds from a small-footprint LiDAR to estimate the orientation parameters of the MMS trajectory. They derived 2D tree stem positions using integrated GNSS/INS and SLAM/INS trajectories to compare the accuracy in open and mature forest areas. The accuracy of the SLAM/INS trajectory was 0.16 m and 0.27 m in the X and Y directions, respectively, as compared with 0.36 m and 0.32 m from GNSS/INS trajectory. Their key limitations are: (a) the approach required the range of points to be limited (25 m in their study) to ensure a small laser beam footprint size; (b) they did not estimate positional trajectory parameters; (c) their approach did not include GNSS with LiDAR and INS to refine the trajectory; and (d) the achieved accuracy was suitable for tree stem localization, but was not adequate for more intricate metrics, such as tree diameter, basal area, etc. Qian et al. [35] used the same platform as Tang et al. [34] in an approach that incorporated heading angles and velocities from GNSS/INS to improve the positional accuracy of LiDAR-based SLAM. Both position and orientation parameters of the trajectory were estimated in their research. They extracted 2D tree stem locations with a 0.06 m accuracy from point clouds reconstructed using the derived trajectory. The main limitation of their approach is the requirement of a very good distribution of features to obtain reliable results for LiDAR-based SLAM. Moreover, while the approach provided good results in feature-rich forests, it did not perform well in open forests with very sparse trees.

Su et al. [26] developed a Backpack MMS composed of two LiDAR units and an Inertial Measurement Unit (IMU). They implemented a LiDAR-based SLAM to estimate the trajectory position and orientation parameters. They derived two forest inventory variables—tree height and DBH. The tree height was estimated with an accuracy of 2.24 m and the DBH had an accuracy of 0.03 m. There are several limitations to their study: (a) their approach relied solely on LiDAR-SLAM, whereas GNSS and INS were not integrated, i.e., the data were not georeferenced, thus rendering it impossible to conduct multi-temporal data acquisition and forest monitoring; (b) their LiDAR-SLAM strategy required a considerable amount of manual correction to generate reasonable LiDAR point clouds in complex natural forests; (c) they did not investigate or address the misalignment of point clouds in the same area captured from different Backpack MMS tracks; and (d) their study sites were relatively small (30 m by 30 m), with about 12–27 trees in each site and, thus, the duration of each data acquisition was very short to be impacted by inaccurate trajectory parameters. Polewski et al. [36] explored the possibility of marker-free co-registration of UAV and Backpack LiDAR point clouds in forests. The UAV data were georeferenced, but the Backpack trajectory was estimated using LiDAR-based SLAM. They estimated the rotation, translation, and scaling parameters between the UAV/Backpack point clouds using individual tree locations as tie points. The limitations to their approach are: (a) they did not investigate the trajectory inaccuracy and its impact on point cloud alignment within the Backpack dataset and (b) they assumed a rigid body transformation between the point clouds from the two systems, which is not applicable for cases with inaccurate trajectory during under-canopy mapping, which variably impacts the point cloud from one area to another.

In summary, the limitations of prior research include: (a) traditional approaches rely on acquired data from expensive manned airborne systems, which cannot be collected frequently to provide data with high temporal resolution; (b) cost-effective UAV-based photogrammetric and LiDAR data are unable to provide fine-resolution forest metrics for individual trees; (c) static terrestrial LiDAR systems are prone to occlusions, and require time-consuming and labor-intensive field surveys; (d) mobile ground LiDAR and photogrammetric systems are affected by GNSS signal outages, which deteriorate the

georeferencing accuracy of derived products; and (e) the synergistic characteristics of UAV and mobile ground mapping systems are not fully explored. Research on improving the quality of mobile terrestrial remote sensing systems under GNSS signal outages is still lacking in terms of (a) partially refining positional or attitude information; (b) requiring extensive preprocessing for deriving suitable features for trajectory enhancement; (c) being incapable of handling different sensing modalities (e.g., images together with 2D and 3D LiDAR units); (d) not taking full advantage of onboard IMUs; (e) limiting the range of acquired data to a few meters; (f) not providing georeferenced inventory metrics that could aid in tracking forest growth from temporal data acquisitions; and (g) being quite complex for scalable implementation.

In response to the majority of stated limitations of state-of-the-art techniques for accurate forest stem-level mapping, this study proposes a system-driven framework capable of conducting system calibration and trajectory enhancement for LiDAR units mounted on UAV/Backpack MMS to generate accurate point clouds for forest inventory. More specifically, features including tree trunks and terrain patches are extracted from the LiDAR point clouds. By minimizing discrepancies among features captured from different timestamps/tracks and different systems while considering both absolute and relative positional/rotational information provided by the GNSS/INS-based trajectory, system calibration and trajectory information are refined through a non-linear least squares adjustment (LSA) process. The key contributions of this work are itemized as follows:

- Develop a general, system-driven framework capable of conducting system calibration and/or trajectory enhancement for LiDAR MMS in forest environments, while deriving forest inventory biometrics such as tree trunk radius and orientation;
- Conduct in situ system calibration and trajectory enhancement for UAV datasets under leaf-off conditions to derive point clouds with high relative and absolute accuracy;
- Assess the performance of the proposed trajectory enhancement strategy for Backpack datasets with trajectories of varying quality in young/mature plantations and examine if UAV data can be used as a reference to improve the relative/absolute quality of Backpack point clouds.

The remainder of this paper is structured as follows: Section 2 introduces the UAV/Backpack MMS, study sites, and acquired datasets used in this study; Section 3 proposes a system-driven strategy for system calibration and trajectory enhancement utilizing terrain patches and tree trunks extracted from LiDAR point clouds; Section 4 presents experimental results for UAV and Backpack datasets to evaluate the performance of the proposed strategy; finally, Section 5 summarizes the findings of the research along with recommendations for future work.

## 2. Acquisition Systems and Dataset Description

For this study, a total of five datasets were collected from two forest plantations using three UAV and one Backpack mobile mapping systems. These systems were developed in-house by the Digital Photogrammetry Research Group (DPRG) at Purdue University. In this section, the four mobile mapping systems are first introduced, followed by a description of the covered plantation areas and acquired datasets.

### 2.1. UAV and Backpack MMS

In this study, three UAV systems were utilized (namely, *UAV-1*, *UAV-2*, and *UAV-3* systems). The *UAV-1* system (Figure 1a) comprised a LiDAR unit—Velodyne VLP-32C [37]—and a camera—Sony $\alpha$7R III. The *UAV-2* system shared the same payload as the *UAV-1* system, except that a Sony $\alpha$7R camera was used. The *UAV-3* system (as shown in Figure 1b) carried an Ouster OS2-64 LiDAR [38], a Sony RX1RII camera, and a Headwall Nano Hyperspec VNIR camera. The LiDAR units on the UAV systems were mounted with their rotation axes approximately parallel to the flying direction. For all systems, the LiDAR data were directly georeferenced through an Applanix APX15 v3 GNSS/INS unit [39]. The GNSS/INS unit, with an IMU data rate of 200 Hz, provided trajectory with an accuracy of

2–5 cm for position, 0.025° for roll/pitch angles, and 0.080° for heading angle in open sky conditions after post-processing. The Backpack MMS (as shown in Figure 1c) consisted of a Velodyne VLP-16 Hi-Res LiDAR [40] and a Sony α7R II camera. LiDAR data were directly georeferenced using a Novatel SPAN-CPT GNSS/INS unit [41]. For this unit, the IMU data rate was 100 Hz, and it provided a post-processing accuracy of 1–2 cm for position, 0.008° for roll/pitch angles, and 0.026° for heading angle.

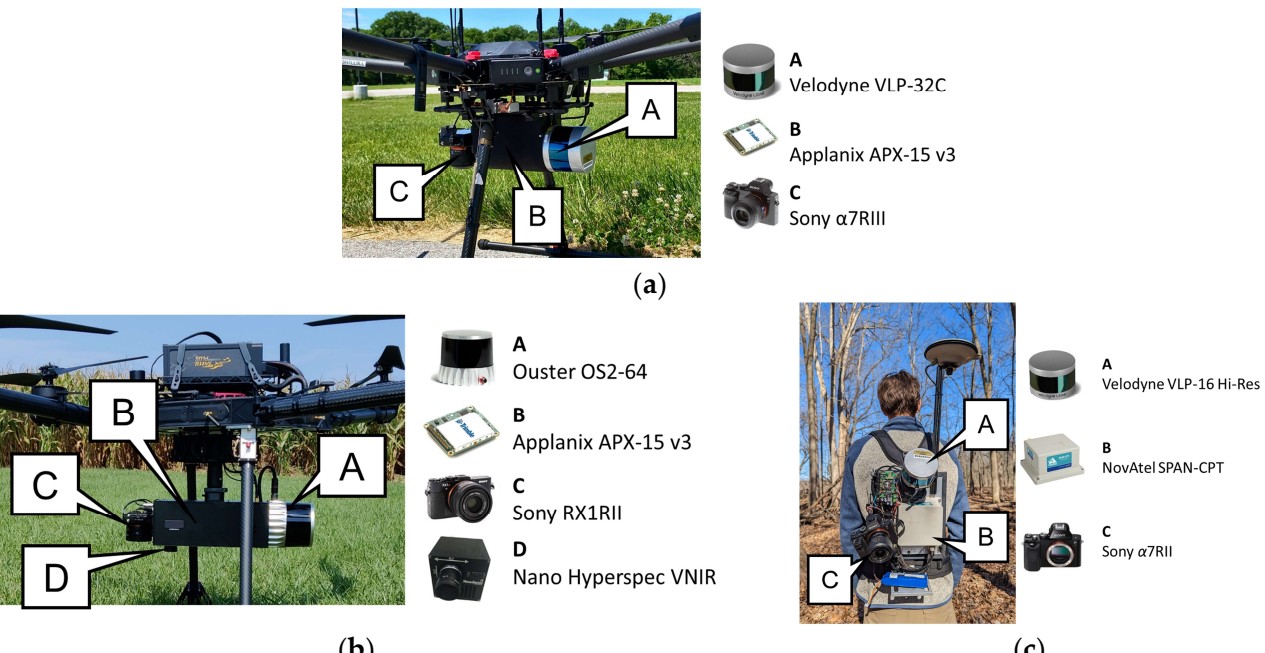

**Figure 1.** Utilized mobile mapping systems and onboard sensors in this study: (**a**) *UAV-1* system, (**b**) *UAV-3* system, and (**c**) Backpack system.

To generate LiDAR point clouds from these MMSs, mounting parameters including lever arm and boresight angles relating the LiDAR unit frame to the GNSS/INS body frame needed to be established. In this study, the UAV and Backpack MMSs underwent a feature-based system calibration [42]. Based on the specifications of the involved sensors, the expected accuracy of the point cloud following the system calibration was estimated using a LiDAR Error Propagation Calculator [43]; for the UAV MMS with 50 m flying height, the expected accuracy was 5–6 cm at the nadir position in both horizontal and vertical directions. The horizontal accuracy increased to 8–9 cm at the edge of the swath. For the Backpack system, the expected accuracy was 3 cm at a range of 50 m.

### 2.2. Study Sites

The study sites used for this research were a young—Plot 115—and a mature—Plot 3b—forest plantation located at Martell Forest, West Lafayette, IN, USA, as shown in Figure 2a. Martell Forest is a research forest owned and managed by Purdue University. Plot 115 was planted in 2007 with northern red oak (*Quercus rubra*) as the primary species and burr oak (*Q. macrocarpa*) as trainers. The plot follows a grid pattern consisting of 22 rows and 50 trees per row. The spacing between neighboring rows is around 5 m and between-tree spacing within a row is approximately 2.5 m. Tree height in the study area ranges from 10 to 13 m at measurement year 13 with an average DBH of 12.7 cm. Within each row, the branches of neighboring trees interlace with each other. The understory vegetation within the plot, including voluntary seedlings and herbaceous species, is removed on an annual basis. In 2021, there were a total of 1,080 trees in Plot 115. The plot went through a tree-thinning activity in late February 2022, and 410 trees were cut down, as shown in Figure 2b. The mature plantation Plot 3b is a northern red oak (*Quercus rubra*) provenance

test that was successively planted in 1962, 1963, and 1964 with seed sources throughout the eastern USA and Canada. The initial planting pattern had 50 rows and 34 columns, and the between-row/between-column distance is approximately 2.5 m. After decades of destructive felling of some trees for research purposes, there are around 550 trees left. Tree height in the study area ranges from 24 to 28 m.

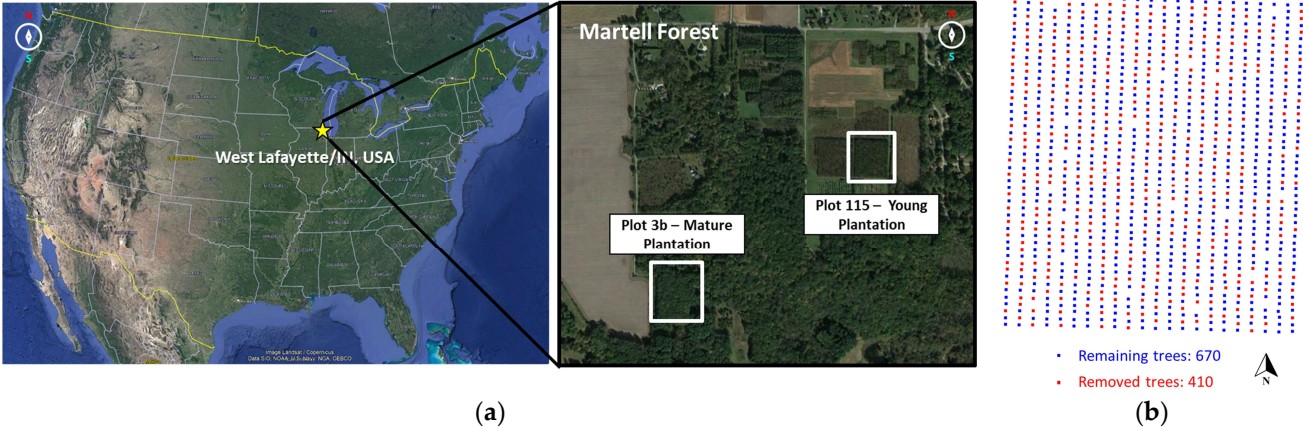

**Figure 2.** Two study sites at Martell Forest (Plot 115 and Plot 3b): (**a**) locations of the study sites and (**b**) removed and remaining trees after the thinning activity in Plot 115 during late February 2022.

### 2.3. Dataset Description

For the young plantation, Plot 115, three datasets were acquired: (a) the *YP-UAV-2021* dataset collected by the *UAV-1* system on 13 March 2021 under leaf-off conditions, (b) the *YP-UAV-2022* dataset collected by the *UAV-2* system on 3 March 2022 under leaf-off conditions, and (c) the *YP-BP-2021* dataset collected by the Backpack system on 5 August 2021 under leaf-on conditions. It is worth noting that the *YP-UAV-2022* dataset was obtained after the tree-thinning activity. As for the mature plantation Plot 3b, two datasets were collected under leaf-off conditions: (a) the *MP-UAV-2023* dataset acquired by the *UAV-3* system on 20 March 2023 and (b) the *MP-BP-2023* dataset acquired by the Backpack system on 7 March 2023. Given that tree trunks might not be captured by UAV LiDAR systems during leaf-on conditions [44], all UAV datasets were collected under leaf-off conditions, as the proposed framework is based on the availability of tree trunk features. To obtain the highest level of accuracy in the UAV-based point clouds, points were reconstructed only when the laser beam pointing direction was within ±70° from the nadir. Further information and details regarding these datasets will be presented in the subsequent subsections.

### 2.3.1. Datasets of the Young Plantation

Both UAV datasets were collected at a 40 m height with a flying speed of 3.5 m/s. The *YP-UAV-2021* mission included 12 east–west flight lines with an 11 m lateral distance between adjacent ones. Considering ±70° off-nadir reconstruction, the side lap of the point cloud was 95%. The *YP-UAV-2022* dataset had 10 flight lines with a 13 m lateral distance, resulting in an approximately 80% side lap. Figure 3 shows a top view of the GNSS/INS-derived trajectory for the two UAV datasets. Both UAV systems were flown above the canopy with continuous access to GNSS signals; thus, the derived trajectory is expected to be accurate. As mentioned earlier, the *YP-UAV-2022* dataset was captured after a tree-thinning activity, resulting in a significant amount of tree debris remaining on the plantation floor. Figure 4a,b present the reconstructed point clouds after height normalization relative to the ground level in the 1–3 m range for the two UAV datasets, respectively. While the *YP-UAV-2021* dataset displays well-defined tree trunks, debris is visible in the *YP-UAV-2022* dataset. This is also evident in a terrestrial image (shown in Figure 4c) captured near the data acquisition date of the *YP-UAV-2022* dataset. The presence of woody debris is expected to create difficulties in extracting tree trunks and

terrain patches, which will be employed in the subsequent system calibration and trajectory enhancement process.

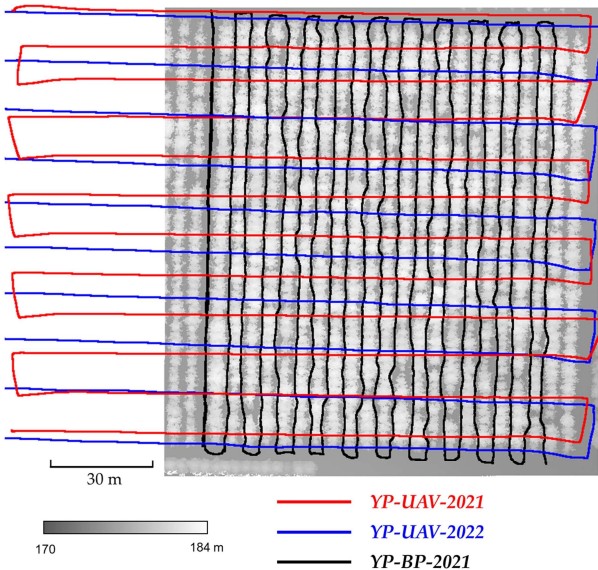

| | | YP-UAV-2021 |
| | | YP-UAV-2022 |
| | | YP-BP-2021 |

**Figure 3.** Top view of the trajectory for the two UAV and one Backpack datasets for young plantation overlaid on the point cloud (colored by height) captured in the *YP-UAV-2021* dataset.

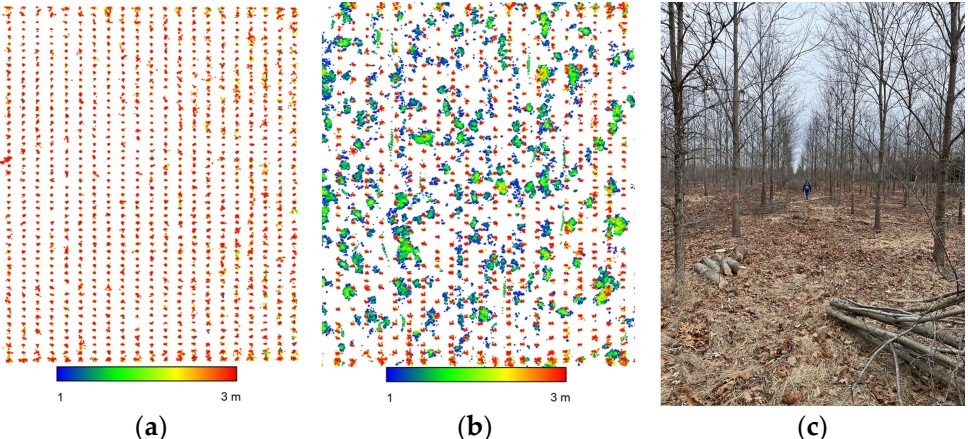

**Figure 4.** Top view of the normalized height point cloud in the 1–3 m range for (**a**) *YP-UAV-2021* and (**b**) *YP-UAV-2022* datasets, as well as (**c**) a terrestrial image showing existing debris captured in the *YP-UAV-2022* dataset.

Furthermore, it is noteworthy that the mounting parameters of the *UAV-2* system are outdated, which resulted in an anticipated decrease in the geometric accuracy for the *YP-UAV-2022* dataset. Figure 5 shows a sample tree from the two UAV datasets where LiDAR points with large range measurements come from flight lines with large planimetric distances to this tree. For the *YP-UAV-2021* point cloud, the noise level in the X direction (along-flight direction) was much higher compared with that in the Y direction (across-flight direction). Noisy points are mostly of large range measurements, thus suggesting issues with the system calibration parameters and/or trajectory that affected the along-flight direction more than the across-flight one. As for the *YP-UAV-2022* dataset with inaccurate mounting parameters, dual versions of the tree trunk can be observed in both the X and Y directions.

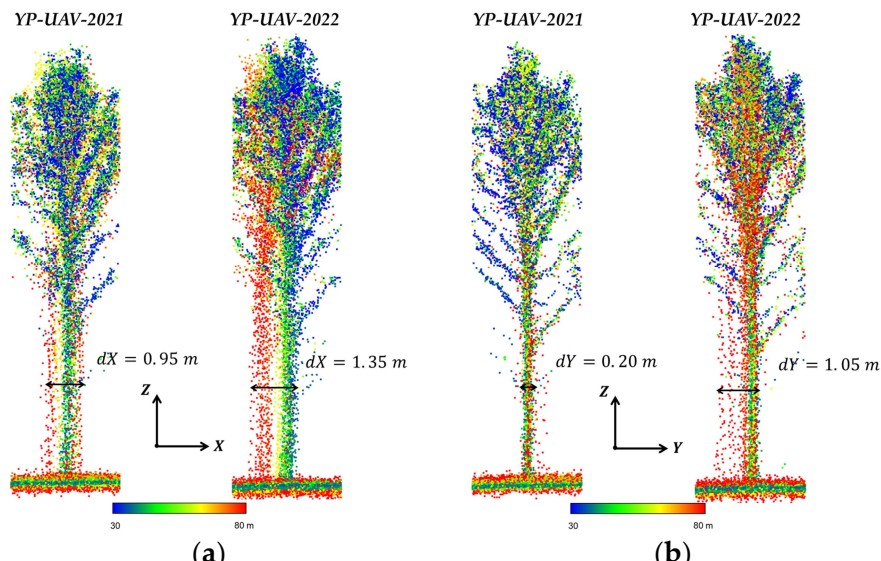

**Figure 5.** A sample tree (colored by LiDAR range) in the *YP-UAV-2021* and *YP-UAV-2022* datasets viewed from the (**a**) X-Z and (**b**) Y-Z planes.

For the *YP-BP-2021* dataset, the Backpack system was carried while walking under the forest canopy between individual tree rows. The mission consisted of 22 north–south tracks and each of them lasted around 2.5 min (as shown in Figure 3). At the end of each track, the operator walked out of the canopy into open sky before the next track. This data acquisition pattern guarantees a trajectory of reasonable quality without a dramatic increase in drifting errors over time. The number of tracked satellites ranged from 3 to 5 and 9 to 11 when walking under and outside the canopy, respectively. To demonstrate the impact of the GNSS signal outages, a small region of interest (ROI) within the young plantation was cropped from the Backpack point cloud generated from the GNSS/INS-derived trajectory, as shown in Figure 6. The ROI chosen for examination is in the middle section of the whole area. A spatial misalignment of about 1.7 m in the horizontal direction and 1.2 m in the vertical direction can be observed in the point cloud.

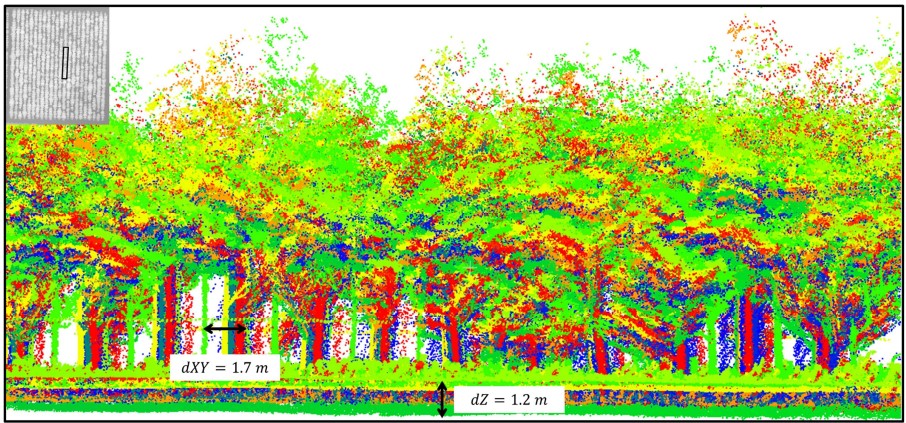

**Figure 6.** Side view of a profile from the *YP-BP-2021* dataset (colored by time) for qualitative evaluation of the level of relative misalignment.

### 2.3.2. Datasets of the Mature Plantation

The *MP-UAV-2023* dataset was acquired by the *UAV-3* system at 50 m above ground height with a speed of 3.6 m/s. The flight mission comprised 14 north–south flight lines, with an 8 m lateral distance between neighboring ones. The side lap percentage of the point cloud was 97% while considering ±70° off-nadir reconstruction. Figure 7 shows a top view

of the UAV trajectory (colored in red) obtained from the GNSS/INS post-processing. In Figure 8a, a sample tree from the UAV dataset is displayed, where point cloud alignment in the X direction is reasonable, except for some noisy points with large range measurements. However, in the Y direction (along-flight direction), the noise level is much higher than that in the X direction.

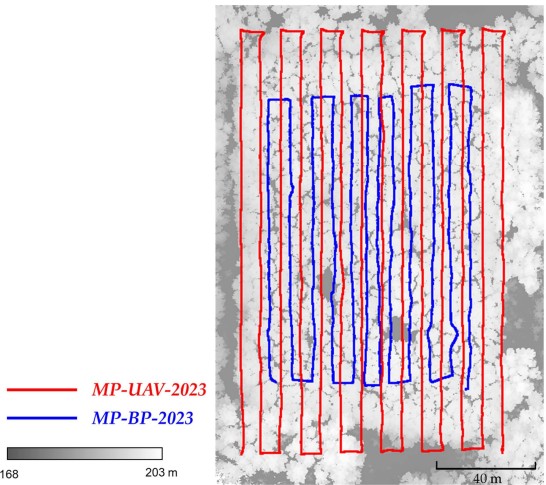

**Figure 7.** Top view of the trajectory for the UAV and Backpack datasets for the mature plantation overlaid on the point cloud (colored by height) captured in the *MP-UAV-2023* dataset.

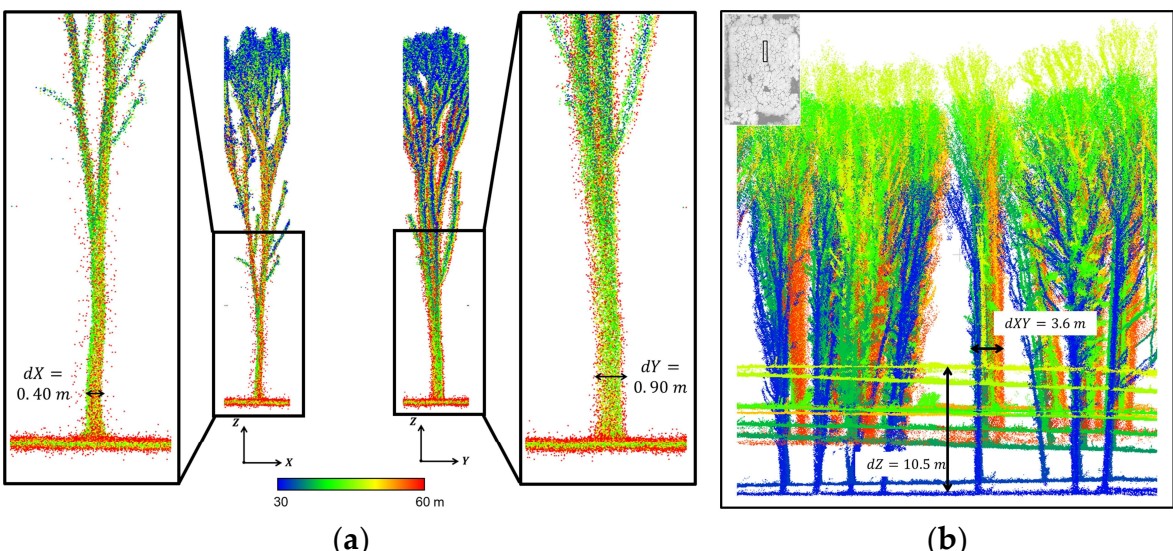

**Figure 8.** (**a**) A sample tree (colored by LiDAR range) in the *MP-UAV-2023* dataset and (**b**) side view of a profile from the *MP-BP-2023* dataset (colored by time) for qualitative evaluation of level of the relative misalignment.

Regarding the *MP-BP-2023* dataset, the mission included 12 north–south tracks and each of them lasted around 2 min, as highlighted in blue in Figure 7. The Backpack system was carried under the canopy throughout the entire mission without leaving the forest, while the number of tracked satellites ranged from 0 to 6. Due to the denser canopy from the mature plantation and lack of access to open sky during the mission, GNSS signal outages are more severe than those for the young plantation. This resulted in a lower quality trajectory of the *MP-BP-2023* dataset compared with that of the *YP-BP-2021* dataset, despite the former being collected in the leaf-off condition while the latter was acquired in the leaf-on condition. An ROI in the middle of this study site was extracted from the Backpack point cloud and is presented in Figure 8b. The misalignment in the horizontal

and vertical directions was up to 3.6 m and 10.5 m, respectively, indicating the poor quality of the derived GNSS/INS trajectory.

## 3. System Calibration and Trajectory Enhancement Strategy

In this study, we propose a system-driven approach to reduce the misalignment within point clouds caused by inaccurate LiDAR mounting parameters and/or GNSS signal outages. The proposed strategy is based on the hypothesis that any inaccuracy related to mounting parameters and/or trajectory information would manifest as discrepancies among conjugate features, as demonstrated in the previous section. Figure 9 depicts the framework for the proposed approach. This approach utilizes common features that can be automatically identified and extracted from point clouds in forest environments. Specifically, terrain patches and tree trunks were derived and used as planar and cylindrical features, respectively, to refine system calibration and trajectory parameters. As shown in Figure 9, Block 1 focused on extracting and matching planar and cylindrical features, while Block 2 showed the optimization framework for system calibration and trajectory enhancement. The output of the optimization process included refined mounting parameters, enhanced trajectory parameters (position and orientation), and estimated feature parameters for each planar/cylindrical primitive. It is worth mentioning that derived parametric models could provide critical forest inventory biometrics such as tree trunk radius and orientation, which is also a key contribution of the proposed strategy.

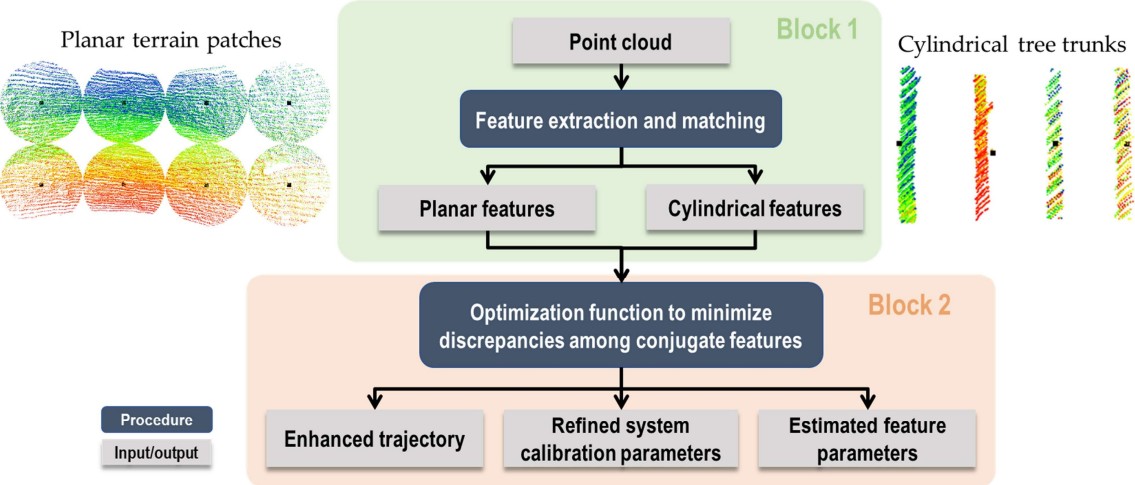

**Figure 9.** Proposed framework for system calibration and trajectory enhancement utilizing terrain patches and tree trunks.

### 3.1. Feature Extraction and Matching

In this subsection, we present the strategy for extracting and matching planar features (terrain patches) and cylindrical features (tree trunks) from point clouds captured by different LiDAR MMSs. A prerequisite for reliable feature extraction is that the used point cloud has a relatively good quality in a local neighborhood. In this study, given that UAV systems have continuous accessibility to GNSS signals, they produced LiDAR point clouds with reasonable quality. Therefore, the feature extraction was directly conducted on the entire point cloud. On the other hand, by assuming that the point cloud from a single track of the Backpack dataset had relatively good quality, feature extraction was performed on each track separately.

To begin with, the point clouds (either the entire point cloud for UAV datasets or individual tracks for the Backpack datasets) were reconstructed using the original GNSS/INS-derived trajectory and initial LiDAR system calibration parameters. Then, a ground filtering algorithm—the adaptive cloth simulation proposed by Lin et al. [45]—was applied to generate a digital terrain model (DTM) and, furthermore, derive bare earth (BE) and

above-ground (AG) points. Specifically, LiDAR points within a height buffer above the DTM (e.g., 0.5 m in this study) were categorized as the BE point cloud, while the remaining points were considered the AG point cloud. The BE point clouds were used for terrain patch extraction and matching, as will be discussed in Section 3.1.1. The AG point clouds were used for conducting individual tree detection/localization followed by tree trunk extraction and matching, as will be discussed in Section 3.1.2.

### 3.1.1. Terrain Patch Extraction and Matching for Vertical Control

Despite the varying size and shape of above-ground forest entities, within a local neighborhood, it is possible to approximate the ground as a planar surface. Terrain patches—i.e., small segments of the BE point cloud in a local area—extracted and matched in individual tracks and/or different datasets, were used as planar features to provide vertical control for system calibration and trajectory enhancement. For terrain patch extraction, regularly spaced 2D seed points were created within the ROI, where the Z coordinates were obtained from the DTM. For each seed point, its neighboring points within a given search radius were identified from the BE point cloud. A dimensionality-based analysis, as described in Demantké et al. [46], was conducted to test the planarity of the local neighborhood. Iterative plane fitting—i.e., multiple iterations of plane fitting followed by removing outliers based on the root-mean-square (RMS) value of the fitting error—was performed to derive segmented points and parameters describing the respective plane model. Figure 10 shows sample terrain patches extracted from one track of the *YP-BP-2021* dataset. Once the terrain patches from different point clouds were extracted, features that met the following criteria were matched: (a) they were extracted from the same seed point and (b) the angle between their normal vectors was within a pre-defined threshold.

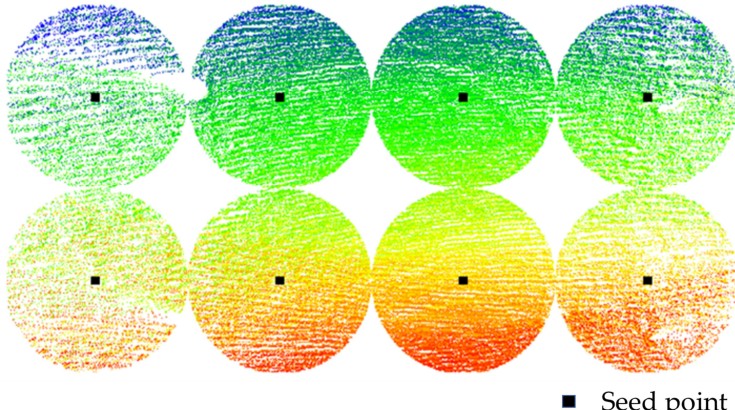

■ Seed point

**Figure 10.** Sample terrain patches derived from the point cloud (colored by time).

### 3.1.2. Tree Trunk Extraction and Matching for Horizontal Control

Tree trunks were used as cylindrical features to provide horizontal control, as they are distinct objects and their planimetric locations remain the same over time. Tree trunk extraction started by isolating the lower portion of the AG point cloud (i.e., hypothesized trunk portion) based on pre-defined height range thresholds (namely, $h_{min}$ and $h_{max}$) above the DTM, as shown in Figure 11. The height/density-based peak detection approach proposed by Lin et al. [44] was adopted to identify individual trees in the hypothesized trunk portion. Their approach utilizes a grid-based evaluation of the sum of normalized elevations relative to the ground level of all points to identify local peaks that will be designated as tree trunk locations. Detected planimetric locations are then adopted to identify corresponding LiDAR points from the hypothesized trunk portion. First, for a detected tree with planimetric location $(X_t, Y_t)$, the seed point that corresponded to the tree trunk portion was defined as $(X_t, Y_t, Z_G + \Delta h)$, where $Z_G$ is the ground height derived from DTM and $\Delta h$ is a pre-defined height above ground ($\Delta h$ is selected to be within the range of $h_{min}$ and $h_{max}$). Then, a spherical region centered at the seed point

with a pre-defined radius (e.g., 0.5 m) was created. LiDAR points within this region of the hypothesized trunk portion were analyzed to determine the presence of a linear/cylindrical feature using a dimensionality-based approach [46]. If a linear/cylindrical feature was detected, an iterative model fitting and outlier removal process was employed to derive the cylindrical parametric model. Subsequently, a region growing approach was applied to incrementally include neighboring points that belong to the current feature, provided their normal distances from the corresponding parametric model were below a specified multiple by the RMS of the fitting error. The augmentation process terminated when no points could be included in this feature. The output of the feature extraction included the segmented points and parameters describing the respective cylinder model. Figure 11 shows sample tree trunks extracted from an individual track of the *YP-BP-2021* dataset.

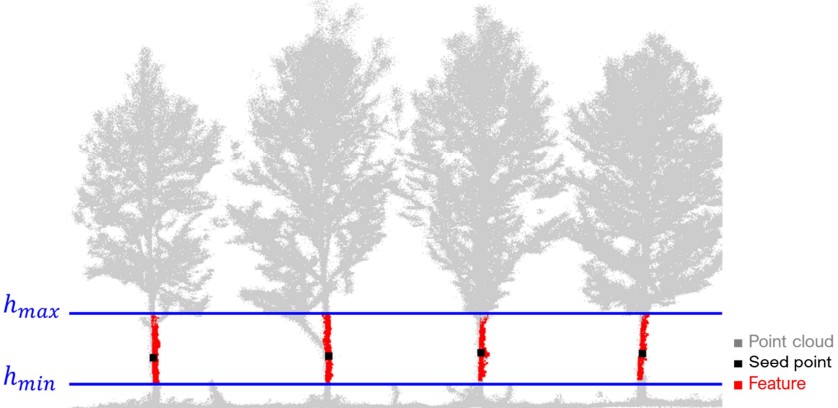

**Figure 11.** Minimum and maximum height thresholds used for tree-trunk extraction and sample cylindrical features (tree trunks) extracted from the point cloud (side view)—seed points for tree trunk segmentation are represented by small black squares.

After extracting tree trunks from each point cloud, a feature matching procedure was carried out to identify conjugate trunks from different point clouds. Since misalignment among the point clouds was most significant in the Z direction, planimetric locations of the extracted tree trunks were utilized in the matching. To illustrate the proposed strategy, we used a Backpack dataset as an example, since tree trunk matching is especially challenging for such data due to GNSS signal outages. Although drifting errors caused by signal outages may increase over time, the relative alignment between successive LiDAR tracks was still of reasonable quality. Thus, the matching process was sequentially performed on successive tracks. Specifically, for a LiDAR track pair, an Iterative Closest Point (ICP) algorithm was used to register the 2D locations of derived tree trunks from the candidate track to those from the reference track. If two trunks from different tracks had a planimetric distance smaller than 0.5 m after the registration, they were considered conjugate. However, this process assumes that the misalignment between two tracks in the planimetric direction can be modeled by a 2D rigid body transformation, which is not always true. To address this issue, derived tree trunks from the candidate track can be divided into several portions along the track. The registration and matching process can then be conducted on each portion individually. Furthermore, if a UAV dataset was included in this procedure, we used its derived 2D trunk locations from the entire point cloud as reference due to their high absolute accuracy. Then, tree trunks from each Backpack track or subtrack were directly matched to the UAV reference ones.

*3.2. Optimization Framework for System Calibration and Trajectory Enhancement*

Extracted and matched planar/cylindrical features from single or multiple datasets were adopted to (a) refine the LiDAR system calibration parameters, (b) enhance the quality of the GNSS/INS-derived trajectory, and (c) improve the alignment of the point cloud. Con-

ceptually, the proposed optimization framework aimed at minimizing the normal distance between LiDAR points and the respective parametric models for planar/cylindrical features by refining system calibration and trajectory parameters through a non-linear LSA. The point positioning equation was the basis of this optimization framework. For any LiDAR point $I$ captured at time $t$, its coordinates in the mapping frame $\left(r_I^m(t)\right)$ were defined by the trajectory position and orientation parameters at the corresponding time $\left(r_{b(t)}^m, R_{b(t)}^m\right)$, LiDAR mounting parameters including lever arm and boresight angles $\left(r_{lu}^b, R_{lu}^b\right)$, and laser unit frame coordinates of the point at the firing time $\left(r_I^{lu(t)}\right)$. $r_I^{lu(t)}$ was derived from raw LiDAR measurements, including range and pointing direction measurements of the LiDAR unit. The mathematical model is expressed symbolically in Equation (1). The corrected coordinates of the same point after system calibration and trajectory enhancement will depend on the refined mounting parameters $\left(r_{lu}^b(\text{refined}), R_{lu}^b(\text{refined})\right)$ and estimated corrections to the trajectory position/orientation parameters $\left(\delta r_{b(t)}^m, \delta R_{b(t)}^m\right)$, as expressed in Equation (2).

$$r_I^m(t) = f\left(r_{b(t)}^m, R_{b(t)}^m, r_{lu}^b, R_{lu}^b, r_I^{lu(t)}\right) \tag{1}$$

$$r_I^m(t)_{\text{corrected}} = f\left(r_{b(t)}^m, \delta r_{b(t)}^m, R_{b(t)}^m, \delta R_{b(t)}^m, r_{lu}^b(\text{refined}), R_{lu}^b(\text{refined}), r_I^{lu(t)}\right) \tag{2}$$

Solving for the trajectory corrections at every timestamp of laser beam firing is not recommended as it would cause over-parametrization in the LSA. Since we were dealing with a platform that exhibited a relatively smooth trajectory and moderate dynamics, the initial high-frequency trajectory (e.g., 100–200 Hz) was reduced to a lower frequency, using a user-defined down-sampling time interval $\Delta T$. The down-sampled trajectory points are henceforth denoted as trajectory reference points, as shown in Figure 12. The trajectory corrections at a particular firing timestamp were then modeled as $p^{th}$-order polynomial functions of unknown corrections for their $n$ neighboring trajectory reference points. Symbolically, this polynomial modeling is expressed in Equation (3), where it can be seen that for a generic timestamp, $T_0$, its trajectory corrections (denoted generically as $\delta\theta_{b(T_0)}^m$) are a function of the polynomial order along with the timestamps and trajectory corrections of its $n$ closest reference points. The selection of the down-sampling time interval $\Delta T$, polynomial order $p$, and number of neighboring trajectory reference points $n$, is determined by considering the characteristics of the platform dynamics.

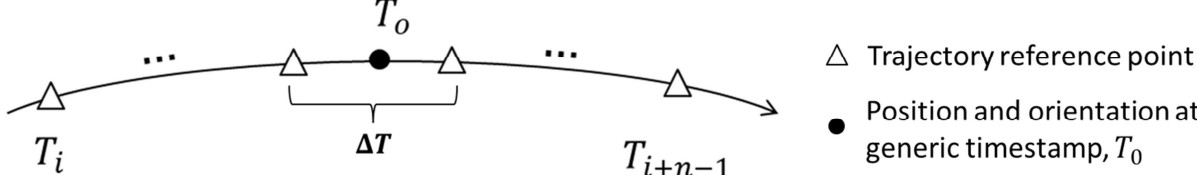

**Figure 12.** Derived trajectory reference points (with a time interval $\Delta T$) from the original high-frequency trajectory: $T_i$ to $T_{i+n-1}$ denote the $n$ closest reference points for a firing timestamp $T_0$.

$$\delta\theta_{b(T_0)}^m = f\left(p, T_0, T_i, T_{i+1}, \dots, T_{i+n-1}, \delta\theta_{b(T_i)}^m, \delta\theta_{b(T_{i+1})}^m, \dots, \delta\theta_{b(T_{i+n-1})}^m\right) \tag{3}$$

The mathematical models for the proposed LSA encompass two types of constraints: (a) objective functions arising from LiDAR feature points and (b) objective functions to incorporate prior information provided by GNSS/INS trajectory. The former aims at minimizing the normal distance of each LiDAR point from the parametric model of its corresponding planar/cylindrical feature. The minimization function is expressed mathematically in Equation (4). Here, $F_k^m$ denotes the parametric model of the $k^{th}$ feature in the mapping frame, and $nd\left(I, t, F_k^m\right)$ represents the normal distance from the LiDAR point $I$ captured at

time $t$ to its corresponding feature $k$. An a priori variance, denoted by $\sigma_{nd}^2$, is associated with each of these constraint equations to assign range-based adaptive weights to the normal distance for each LiDAR feature point. LiDAR feature points captured within a pre-defined range threshold from the LiDAR unit ($\rho_{max}$) are assumed to conform to their corresponding features more accurately than further points, which would be more prone to noise. The adaptive variance is defined according to Equation (5), which describes the assumption that a point with a range $\rho_i$ below $\rho_{max}$ exhibits a constant variance of $\sigma_{ref}^2$, while a point with a greater range has an increasing variance (or, lesser weight). Here, $\sigma_{ref}^2$ represents the nominal expected variance for the LiDAR points and can be determined from the error propagation according to the sensors' specifications. The parameters describing a planar feature include the normal vector $(w_x, w_y, w_z)$ and its normal distance from the origin $d$, as shown in Figure 13a; out of these four parameters, three can be designated as independent. In this work, $w_z$ is fixed to one since the normal vectors of all terrain patches will have a predominant component along the Z-axis. The parameters for a cylindrical feature include the direction vector of its axis $(u_x, u_y, u_z)$, a point on the axis $(x_0, y_0, z_0)$, and radius $r$, as shown in Figure 13b. Out of these parameters, there are only five independent ones—two out of $(u_x, u_y, u_z)$, two out of $(x_0, y_0, z_0)$, and $r$. Since the orientation of all tree trunks will be predominantly vertical, in this study, $u_z$ and $z_0$ are fixed to 1 and 0, respectively.

$$\underset{ff_{b(T_{ref})}^m, r_{lu}^b, R_{lu}^b, F_k^m}{\operatorname{argmin}} \sum_{\forall \text{LiDAR feature points}} \frac{\left(nd\left(I, t, F_k^m\right)\right)^2}{\sigma_{nd}^2} \tag{4}$$

$$\sigma_{nd}^2 = \left(\frac{\max(\rho_{max}, \rho_i)}{\rho_{max}} \times \sigma_{\text{ref}}\right)^2 \tag{5}$$

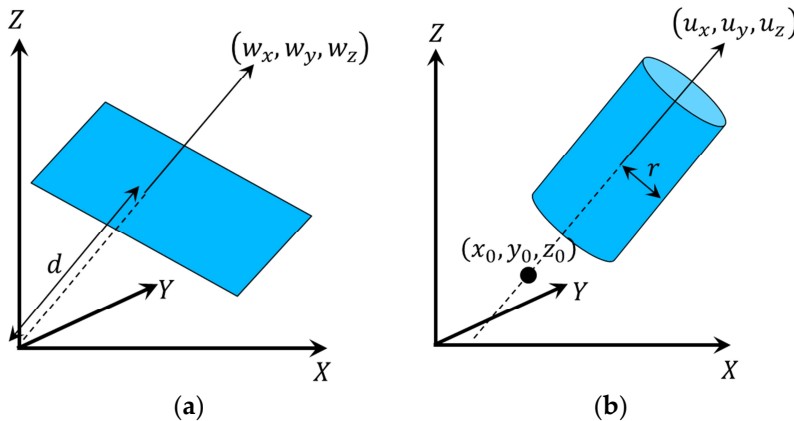

**Figure 13.** Geometric representation of feature parameters: (**a**) planar features and (**b**) cylindrical features.

A second set of constraint equations was introduced in the LSA to ensure that the corrections to the trajectory reference points were commensurate with provided information by the GNSS/INS post-processing from absolute and relative points of view. In an absolute sense, constraint equations were introduced to minimize the change in position and orientation parameters $\delta\theta_{b(t)}^m$ for each trajectory reference point depending on the reported standard deviation by the GNSS/INS post-processing, as given by Equation (6), where $N_t$ denotes the total number of trajectory reference points. From a relative point of view, the change in the distance traversed between two consecutive trajectory reference points was minimized based on the reported velocity accuracy of the trajectory, as given by Equation (7). In this equation, $D_i$ denotes the distance between the positions of the $i^{th}$ and $(i+1)^{th}$ reference points. By including these constraint equations, the short-term (mainly provided by the IMU) and long-term (mainly provided by the GNSS) information from the GNSS/INS-based trajectory were utilized together with the LiDAR observations to ensure

the best accuracy of the enhanced trajectory. One should note that adding the minimization constraints in Equations (6) and (7) based on prior knowledge ensured that corrections to trajectory reference points were zero if there were no LiDAR feature points to assist in their estimation (i.e., there were no feature points with timestamps associated with these reference points).

$$\operatorname*{argmin}_{\delta\theta^m_{b(t)_i}} \sum_{i=1}^{N_t} \frac{\left(\delta\theta^m_{b(t)_i}\right)^2}{\sigma^2_{\theta_i}} \tag{6}$$

$$\operatorname*{argmin}_{\delta r^m_{b(t)_i}, \delta r^m_{b(t)_{i+1}}} \sum_{i=1}^{N_t-1} \frac{\left(D_{i\,\text{corrected}}\left(r^m_{b(t)_i}, \delta r^m_{b(t)_i}, r^m_{b(t)_{i+1}}, \delta r^m_{b(t)_{i+1}}\right) - D_i\left(r^m_{b(t)_i}, r^m_{b(t)_{i+1}}\right)\right)^2}{\sigma^2_v} \tag{7}$$

Based on the above discussion, we can determine the total number of unknowns and constraint equations involved in the LSA model for dataset(s) with $N_{Lu}$ LiDAR units, $N_t$ trajectory reference points, $np_p$ LiDAR points captured over $N_p$ planar terrain patches, and $np_c$ LiDAR points captured over $N_c$ cylindrical tree trunks. The unknowns include $6N_{Lu}$ LiDAR mounting parameters, $6N_t$ trajectory reference point corrections to position/orientation parameters ($\delta\theta$), $3N_p$ planar terrain patch feature parameters (one for each planar feature is fixed), and $5N_c$ cylindrical tree trunk feature parameters (two for each cylindrical feature are fixed). It is worth mentioning that due to the potential correlation between LiDAR mounting parameters and trajectory information, conducting system calibration and trajectory enhancement simultaneously could lead to inaccurate estimation of the involved parameters. Therefore, in the case that both tasks are required, system calibration will be performed while fixing the trajectory information, followed by trajectory enhancement while fixing the refined mounting parameters.

The estimation of the system calibration parameters and trajectory corrections is based on the contribution of the LiDAR constraints (Equation (4)). The effective contribution towards LSA is only along the normal direction(s) to the feature [47]. Hence, there is an effective contribution of one equation per LiDAR point captured over a planar or cylindrical feature, thus resulting in a total of $np_p$ and $np_c$ constraint equations from LiDAR points along terrain patches and tree trunks, respectively. Additionally, there are $6N_t$ and $(N_t - 1)$ constraint equations based on prior absolute and relative trajectory information, respectively. The LSA for system calibration and trajectory enhancement is conducted iteratively until the change in the RMS of normal distances of the LiDAR points from the corresponding features is less than a pre-defined threshold. Once the trajectory enhancement is conducted, estimated corrections of reference points are then used to correct the initial trajectory information at the original data rate through the utilized $p^{th}$-order polynomial function.

The proposed system calibration and trajectory enhancement strategy, including feature extraction, matching, and optimization framework, offers several salient features. First, the constraint equations are not restricted to a single platform/sensor/dataset. The LSA model can be used to conduct simultaneous multi-sensor, multi-temporal, and multi-platform system calibration and trajectory enhancement. Second, the proposed strategy could be used for refining system calibration parameters and enhancing the trajectory information for one or more systems without including any reference system with accurate trajectory (e.g., UAV datasets in this study). Third, the approach allows using one or more systems with accurate system calibration parameters and trajectories as a reference to refine the respective parameters for other systems. In order to accomplish this, the system calibration parameters and trajectory parameters for the reference system(s) are fixed (or assigned low a priori variance) in the LSA model while the parameters for other systems are estimated. For instance, in this study, the UAV MMS dataset with relatively accurate trajectory information could be used as a reference while conducting trajectory enhancement for the Backpack MMS dataset with poor-quality trajectory. Lastly, some

critical forest inventory biometrics such as tree trunk radius and orientation are also derived in the proposed strategy.

## 4. Experimental Results

This section presents experimental results to validate the proposed system calibration and trajectory enhancement strategy in terms of the improvement in the alignment of UAV and Backpack MMS point clouds. It is worth mentioning that the mounting parameters of the Backpack system were calibrated and the sensor-to-object distances during the data acquisition were quite close. In this case, the impact of potential errors in the mounting parameters on the 3D coordinates of LiDAR points was negligible compared to that caused by the inaccurate trajectory. Therefore, trajectory enhancement was the focus for the Backpack datasets, while the mounting parameters were assumed errorless in the conducted experiments.

Since the alignment of UAV point clouds was reasonable, terrain patches and tree trunks were extracted from the entire point cloud from each UAV dataset. Therefore, there was no need for intra-dataset feature matching. The used radius for the extraction of planar terrain patches was set to 1 m. For tree trunk extraction, the minimum and maximum height thresholds ($h_{min}$ and $h_{max}$) were set to 0.5 m and 2.5 m for the *YP-UAV-2021* dataset. Due to existing debris in the *YP-UAV-2022* dataset and understory vegetation in the mature plantation for the *MP-UAV-2023* dataset, the respective height range thresholds were set to 1.5 m and 3.5 m. A total of 3248 terrain patches were extracted from each of the UAV datasets over the young plantation, while 843 and 540 tree trunks were identified in the *YP-UAV-2021* and *YP-UAV-2022* datasets, respectively. The fewer tree trunks in the *YP-UAV-2022* dataset were due to the thinning activity conducted prior to this acquisition. As for the *MP-UAV-2023* dataset, 531 tree trunks and 4610 terrain patches were derived.

For the *YP-BP-2021* and *MP-BP-2023* datasets, the individual tracks were reconstructed using the GNSS/INS trajectory. Then, feature extraction was conducted on each track separately. The feature extraction parameters for the two Backpack datasets were set to be identical with those for the *YP-UAV-2021* and *MP-UAV-2023* datasets, respectively. In total, 3248/3260 terrain patches and 929/499 tree trunks were established for the *YP-BP-2021/MP-BP-2023* datasets. Since this study also aims at investigating the impact of incorporating reference UAV for trajectory enhancement for Backpack datasets with poor-quality trajectories, features from the *MP-BP-2023* and *MP-UAV-2023* datasets were matched, resulting in 3256 and 492 common terrain patches and tree trunks, respectively. One should note that unmatched features between these datasets were still used in the LSA, as they still contributed towards the estimation of dataset/system-specific parameters.

The expected accuracy of post-LSA normal distance related to the planar and cylindrical features ($\sigma_{\text{ref}}$) was set to 5 cm. While conducting trajectory enhancement for UAV and Backpack datasets, the trajectory reference points were established at a frequency of 1 Hz (the frequency of the original trajectory was 200 Hz and 100 Hz for the UAV and Backpack systems, respectively). Corrections to the position and orientation parameters at the laser beam firing timestamps were established using those associated with the three neighboring reference points through a second-order polynomial function. The performance of the proposed system calibration and trajectory enhancement approach was evaluated as follows:

**Estimated trajectory corrections**: The evaluated corrections for the high-frequency trajectory (i.e., following the interpolation process while using estimated corrections for the reference points) were used to illustrate the required trajectory changes to ensure better alignment for the point cloud. Statistical measures (mean and STD) and magnitude of the corrections for individual poses were reported in a tabular form and visualized using a color-coded trajectory with the colors representing the magnitude of applied corrections.

**Relative accuracy of derived point clouds**: The relative accuracy was qualitatively assessed by checking the alignment of the point cloud in an individual dataset corresponding to a profile and/or individual trees. For quantitative assessment, statistical measures of

normal distances between the LiDAR points and their respective best-fitting plane/cylinder before and after the LSA process were reported.

**Absolute accuracy of derived point clouds**: Since the two UAV datasets over the young plantation were collected in different years using different systems, well-aligned point clouds from such datasets indicate that the conducted system calibration and trajectory enhancement framework achieved high absolute accuracy for all acquired UAV datasets. Then, results from the UAV datasets were used as references to analyze the absolute accuracy of the Backpack point clouds after trajectory enhancement for the young and mature plantations. The above comparison was performed qualitatively and quantitatively. The former was conducted by visually checking the alignment of point clouds from different datasets for a profile and/or individual trees. The latter utilized the refined parametric model of extracted/matched terrain and tree trunk features for numerical evaluation of the quality in the Z and X/Y directions, respectively. More specifically, the X and Y coordinates of established seed points for terrain patch extraction were used to derive the Z coordinates from the respective refined plane parameters for each dataset. The differences between the Z values for each terrain patch represented the alignment degree in the vertical direction. For tree trunk features, a point on the refined cylinder axis was derived by setting a common Z coordinate for each dataset (e.g., the Z coordinate of the seed point used for tree trunk extraction from the reference dataset). The derived X and Y coordinates of that point were regarded as the planimetric tree location. The absolute accuracy in the X and Y directions was then estimated using the planimetric distances between respective tree locations from different datasets.

The following discussion starts by presenting the sequential system calibration and trajectory enhancement results for the UAV datasets in Section 4.1; trajectory enhancement results for the Backpack datasets with/without UAV point cloud as a reference will be introduced in Section 4.2.

### 4.1. System Calibration and Trajectory Enhancement for UAV Datasets

In this study, sequential system calibration and trajectory enhancement was conducted on each UAV dataset separately. More specifically, using extracted features, corrections to trajectory reference points were set to zero and fixed while estimating the system calibration parameters in the LSA (due to the acquisition under open sky conditions, observed misalignments were initially attributed to erroneous system calibration parameters). Upon convergence, the refined mounting parameters were fixed and then trajectory corrections were estimated in a second LSA round. The resulting point clouds were finally checked for any additional improvement.

In the first LSA process, LiDAR boresight angles ($\Delta\omega$, $\Delta\phi$, $\Delta\kappa$), as well as lever arm components in the X and Y directions ($\Delta X$ and $\Delta Y$), were estimated. The Z lever arm component was fixed due to the unavailability of vertical control for these datasets [48]. Table 1 presents the initial and refined system calibration parameters, along with their corresponding STD values. The estimated parameters exhibit small STD values, and it has been observed that they are not highly correlated. These findings indicate that LiDAR mounting parameters were accurately estimated. Additionally, by comparing the refined mounting parameters with the initial ones in Table 1, one can see that the lever-arm components remain relatively stable for the three systems. The boresight pitch ($\Delta\phi$) and heading ($\Delta\kappa$) angles of the *UAV-1* system exhibit a change of 0.10°, whereas the boresight heading ($\Delta\kappa$) angles of the *UAV-2* and *UAV-3* systems exhibit a change of around 0.25°. For the *YP-UAV-2022* dataset with 40 m flying height, a change of 0.25° in the heading angle would result in an up to 0.5 m variation in the along-flight direction (X direction) for a LiDAR point at the edge of the swath (i.e., at ±70° off-nadir). Similarly, this change would lead to a 0.6 m variation for the *MP-UAV-2023* dataset at 50 m flying height in the Y (i.e., along-flight) direction.

**Table 1.** Initial and refined mounting parameters using the proposed system calibration approach for the UAV datasets.

| | Mounting Parameters | $\Delta\omega(°)$ | $\Delta\phi(°)$ | $\Delta\kappa(°)$ | $\Delta X(m)$ | $\Delta Y(m)$ | $\Delta Z(m)$ |
|---|---|---|---|---|---|---|---|
| *YP-UAV-2021* | Initial | 0.499 | −0.132 | −0.092 | −0.140 | 0.036 | 0.000 |
| | Refined | 0.466 ±0.001 | −0.249 ±0.002 | −0.193 ±0.003 | −0.133 ±0.001 | 0.042 ±0.001 | N/A |
| *YP-UAV-2022* | Initial | 1.261 | −0.276 | 0.129 | −0.115 | 0.022 | 0.100 |
| | Refined | 1.202 ±0.001 | −0.295 ±0.002 | −0.139 ±0.003 | −0.095 ±0.001 | 0.010 ±0.001 | N/A |
| *MP-UAV-2023* | Initial | 0.392 | 0.077 | 0.022 | −0.042 | −0.039 | 0.014 |
| | Refined | 0.364 ±0.001 | 0.096 ±0.002 | 0.286 ±0.002 | −0.053 ±0.001 | −0.045 ±0.001 | N/A |

To evaluate the improvement after system calibration, Figure 14 displays sample trees from the three UAV datasets using the initial (in red) and refined (in blue) mounting parameters. The figure shows that misalignment in the *YP-UAV-2021* dataset decreased slightly after the system calibration. However, the level of alignment in the X direction was still worse than that in the Y direction. This observation suggests that trajectory enhancement may still be required to achieve a better-quality point cloud. For the *YP-UAV-2022* dataset, the alignment improved more significantly in both the X and Y directions. In terms of the *MP-UAV-2023* dataset, initial point cloud alignment in the Y direction was poorer. After system calibration, those few points with large X discrepancies were closer to the tree trunk, and the alignment in the Y direction improved significantly.

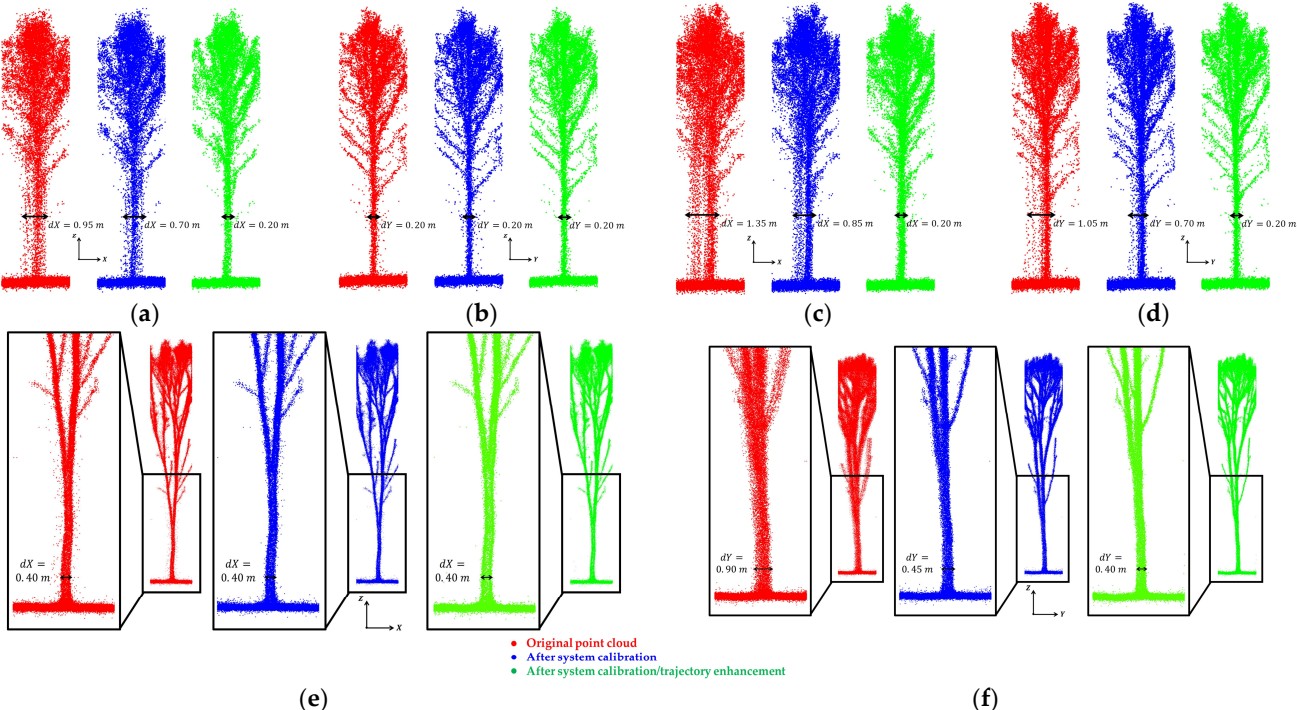

**Figure 14.** A sample tree in the original point clouds (in red), point clouds after system calibration (in blue), as well as point clouds after conducting system calibration and trajectory enhancement (in green): *YP-UAV-2021* dataset views along (**a**) X-Z and (**b**) Y-Z planes, *YP-UAV-2022* dataset views along (**c**) X-Z and (**d**) Y-Z planes, as well as *MP-UAV-2023* dataset views along (**e**) X-Z and (**f**) Y-Z planes.

　　　Once the trajectory enhancement was conducted through a second LSA round while fixing previously estimated mounting parameters, refined trajectory information at the original data rate (200 Hz) was derived. Table 2 presents the mean and STD of the differences between initial and refined position/orientation parameters for the UAV datasets. These statistics were derived while considering only trajectory epochs associated with the used calibration/trajectory enhancement primitives (hereafter denoted as adjusted trajectory epochs). The mean difference values for all parameters were close to zero for the three datasets. The STD values for the position differences were within 4 cm, which was at the same level as the nominal positional accuracy of the GNSS/INS unit. In terms of the orientation parameters, the heading angle ($\kappa$) had the largest STD value (highlighted in bold in Table 2), which can be explained by the relatively lower accuracy of trajectory heading when compared with the roll/pitch angles provided by the GNSS/INS integration. The impact of an inaccurate heading angle on LiDAR points was along the flying direction. This level of heading correction explains the origin of the previously observed worse alignment in the X direction (along-flight direction) in Figure 14a,c for the two datasets over the young plantation. This was confirmed through the improved quality along the X direction for the sample tree in the same figure after conducting the sequential system calibration and trajectory enhancement (in green). Improvements can also be observed in the Y direction for the *YP-UAV-2022* dataset. Regarding the *MP-UAV-2023* dataset, although the improvement was not as significant as the other two shown in Figure 14, the definition of tree trunk/branches was cleaner after refining the trajectory information. In general, after sequential system calibration and trajectory enhancement, the levels of alignment in the X and Y directions for the sample trees of all datasets were quite similar.

**Table 2.** Mean and STD of differences between initial and refined trajectory information for the UAV datasets.

| | Number of Adjusted Trajectory Epochs | Statistics Measures | $X_{dif}$ (m) | $Y_{dif}$ (m) | $Z_{dif}$ ($m$) | $\omega_{dif}$ (°) | $\phi_{dif}$ (°) | $\kappa_{dif}$ (°) |
|---|---|---|---|---|---|---|---|---|
| *YP-UAV-2021* | 95,088 (200 Hz) | Mean | −0.002 | −0.001 | 0.013 | 0.001 | 0.001 | 0.015 |
| | | STD | 0.042 | 0.018 | 0.027 | 0.043 | 0.058 | **0.159** |
| *YP-UAV-2022* | 102,927 (200 Hz) | Mean | −0.001 | 0.000 | 0.004 | 0.004 | 0.001 | −0.046 |
| | | STD | 0.031 | 0.022 | 0.040 | 0.034 | 0.061 | **0.138** |
| *MP-UAV-2023* | 149,588 (200 Hz) | Mean | 0.001 | 0.000 | −0.003 | 0.007 | −0.001 | −0.067 |
| | | STD | 0.042 | 0.043 | 0.032 | 0.067 | 0.050 | **0.109** |

　　　The performance of the proposed system calibration and trajectory enhancement is evaluated quantitatively through Table 3, which reports the mean, STD, and RMS values of normal distances between the LiDAR feature points and their corresponding best-fitting plane/cylinder before and after the two-step LSA. The RMS of normal distances before the LSA indicates that the initial alignment for the *YP-UAV-2021* dataset was better than that of the *YP-UAV-2022* and *MP-UAV-2023* datasets (this was mainly due to the inaccurate boresight heading ($\Delta\kappa$) angles for the latter). After the LSA, significant improvements in the alignment of tree trunks can be observed for all UAV datasets. The RMS value of the normal distances associated with cylindrical features for the *YP-UAV-2021* dataset was 2 cm smaller than that for the *YP-UAV-2022* dataset. This difference can be attributed to the higher height range (i.e., 1.5 m to 3.5 m) used for extracting tree trunk features in the latter dataset to avoid the inclusion of existing debris within the tree trunk features. As a result, more LiDAR points along tree branches were mistakenly extracted as tree trunks, resulting in a larger point to cylindrical feature normal distance. The mature plantation dataset had the smallest residuals for the cylindrical features. This could be explained by the clearer definition of trunks in mature trees compared with young ones, which allows for a more reliable estimation of cylindrical models and leads to smaller fitting errors.

**Table 3.** Quantitative evaluation of point cloud alignment before and after sequential system calibration and trajectory enhancement for the UAV datasets.

| Dataset | Point-to-Feature Normal Distance | # Points (Thousands) | Before LSA | | | After LSA | | |
|---|---|---|---|---|---|---|---|---|
| | | | Mean (m) | STD (m) | RMS (m) | Mean (m) | STD (m) | RMS (m) |
| *YP-UAV-2021* | Planar Features | 10,313 | 0.036 | 0.037 | 0.052 | 0.032 | 0.033 | 0.046 |
| | Cylindrical Features | 412 | 0.107 | 0.106 | 0.151 | 0.048 | 0.053 | 0.072 |
| *YP-UAV-2022* | Planar Features | 10,698 | 0.056 | 0.054 | 0.078 | 0.038 | 0.041 | 0.056 |
| | Cylindrical Features | 310 | 0.181 | 0.147 | 0.233 | 0.061 | 0.076 | 0.097 |
| *MP-UAV-2023* | Planar Features | 6341 | 0.034 | 0.032 | 0.047 | 0.026 | 0.026 | 0.036 |
| | Cylindrical Features | 681 | 0.211 | 0.122 | 0.244 | 0.041 | 0.050 | 0.064 |

The above evaluation focuses on the relative accuracy of each UAV dataset. The absolute accuracy of the point clouds after the proposed system calibration and trajectory enhancement was validated by analyzing the agreement of point clouds from the two datasets over the young plantation. Figure 15 shows the sample tree after sequential system calibration and trajectory enhancement for the *YP-UAV-2021* and *YP-UAV-2022* datasets. It is clear in this figure that the tree trunk and branches aligned well in both the X and Y directions. The alignment in the Z direction was slightly worse than that in the X and Y directions (around an 8 cm Z-shift between the two datasets). This is mainly because the lever arm Z components of the two UAV systems were derived through manual measurements and fixed in the LSA. Moreover, lower alignment quality in the Z direction could be attributed to falling leaves and debris on the plantation floor. Table 4 presents the quantitative evaluation of the point cloud alignment using the extracted features. In the vertical direction, there was a shift of 10 cm between the two point clouds, which is in agreement with the observed discrepancies in Figure 15. For the derived tree trunks, the mean, STD, and RMS values of the X and Y coordinate differences as well as the planimetric distances between tree locations suggest that the tree locations are in agreement with an accuracy of 0.1 m. This planimetric alignment is slightly worse than what has been observed through visual check. Through closer inspection of the point clouds, the large planimetric distances between conjugate tree trunks were mainly caused by the mistakenly extracted branches in the *YP-UAV-2022* dataset. Overall, it can be concluded that after conducting the LSA using the proposed framework, the point clouds from the UAV systems achieved high relative and absolute accuracy. Hereafter, refined point clouds from the *YP-UAV-2021* and *MP-UAV-2023* datasets are used as references to evaluate the performance of trajectory enhancement for the Backpack datasets, which will be presented in the next subsection.

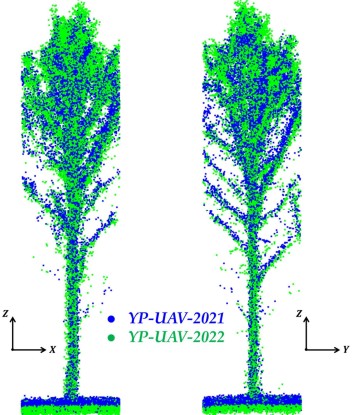

**Figure 15.** A sample tree from the point clouds after sequential system calibration and trajectory enhancement for the *YP-UAV-2021* (in blue) and *YP-UAV-2022* (in green) datasets along the X-Z and Y-Z planes.

**Table 4.** Quantitative evaluation of point cloud alignment before and after sequential system calibration and trajectory enhancement for the UAV datasets.

| Comparison | Statistics Measures | Terrain Patches (3248 Features) | | Tree Trunks (494 Features) | |
|---|---|---|---|---|---|
| | | *dZ* (m) | *dX* (m) | *dY* (m) | Planimetric Distance (m) |
| YP-UAV-2021 vs. YP-UAV-2022 | Mean | −0.099 | 0.019 | −0.059 | 0.097 |
| | STD | 0.037 | 0.055 | 0.089 | 0.073 |
| | RMS | 0.106 | 0.058 | 0.107 | 0.121 |

*4.2. Trajectory Enhancement Results for the Backpack Datasets*

To evaluate the performance of the proposed approach in handling trajectories with different qualities, the two Backpack datasets collected under different conditions were used for the assessment. In this study, the conducted experiments are as follows:

- *YP-BP-2021*: Given that trajectory with frequent access to open sky areas was of reasonable quality, trajectory enhancement was conducted on the *YP-BP-2021* dataset without including the UAV data.
- *MP-BP-2023*: Due to the extended periods of GNSS signal outages, the GNSS/INS-derived trajectory was of lower quality. For this dataset, trajectory enhancement was first conducted using solely Backpack LiDAR (Test 1). Then, the *MP-UAV-2023* data were included as a reference for trajectory enhancement of the Backpack dataset (Test 2). In this test, LiDAR features from the Backpack and UAV datasets were simultaneously included in the LSA, while refined mounting parameters and trajectories for the UAV dataset were treated as errorless.

The relative accuracy of the Backpack point clouds after trajectory enhancement was evaluated first. Figure 16 shows the same profiles (as presented in Section 2.3) from the *YP-BP-2021* and *MP-BP-2023* datasets after trajectory enhancement for the conducted experiments. Compared with the generated point clouds using the GNSS/INS-derived trajectory (as shown in Figures 6 and 8b), drastic improvements can be observed in the Backpack point clouds after trajectory enhancement for all experiments. The initial misalignment of 1–2 m and 10 m for the young/mature plantation point clouds was significantly reduced, where the definition of tree trunks and branches is quite clear. Additionally, the inclusion of the reference UAV dataset did not considerably impact the relative accuracy of the point cloud of the *MP-BP-2023* dataset after trajectory enhancement.

The above findings were further verified through a quantitative evaluation, which is presented in Table 5. This table reports the mean, STD, and RMS values of normal distances of the LiDAR points to their corresponding best-fitting plane/cylinder before and after the LSA process for the three experiments. Before trajectory enhancement, the RMS values indicate that the initial Backpack point cloud alignment for the *YP-BP-2021* dataset was much better than that for the *MP-BP-2023* dataset. After trajectory enhancement, the RMS values for the *YP-BP-2021* dataset show an overall accuracy of 3.4 cm and 2.4 cm for planar and cylindrical features, respectively. The RMS values for the two tests of the *MP-BP-2023* dataset were in the range of 4–5 cm, which were larger due to the presence of low branches and understory vegetation in the mature plantation site. Overall, it can be concluded that all tests achieved high relative accuracy for both datasets. Furthermore, regardless of the trajectory quality, the proposed trajectory enhancement approach allows for deriving point clouds with good intra-dataset alignment using solely Backpack LiDAR data without the need for a reference point cloud.

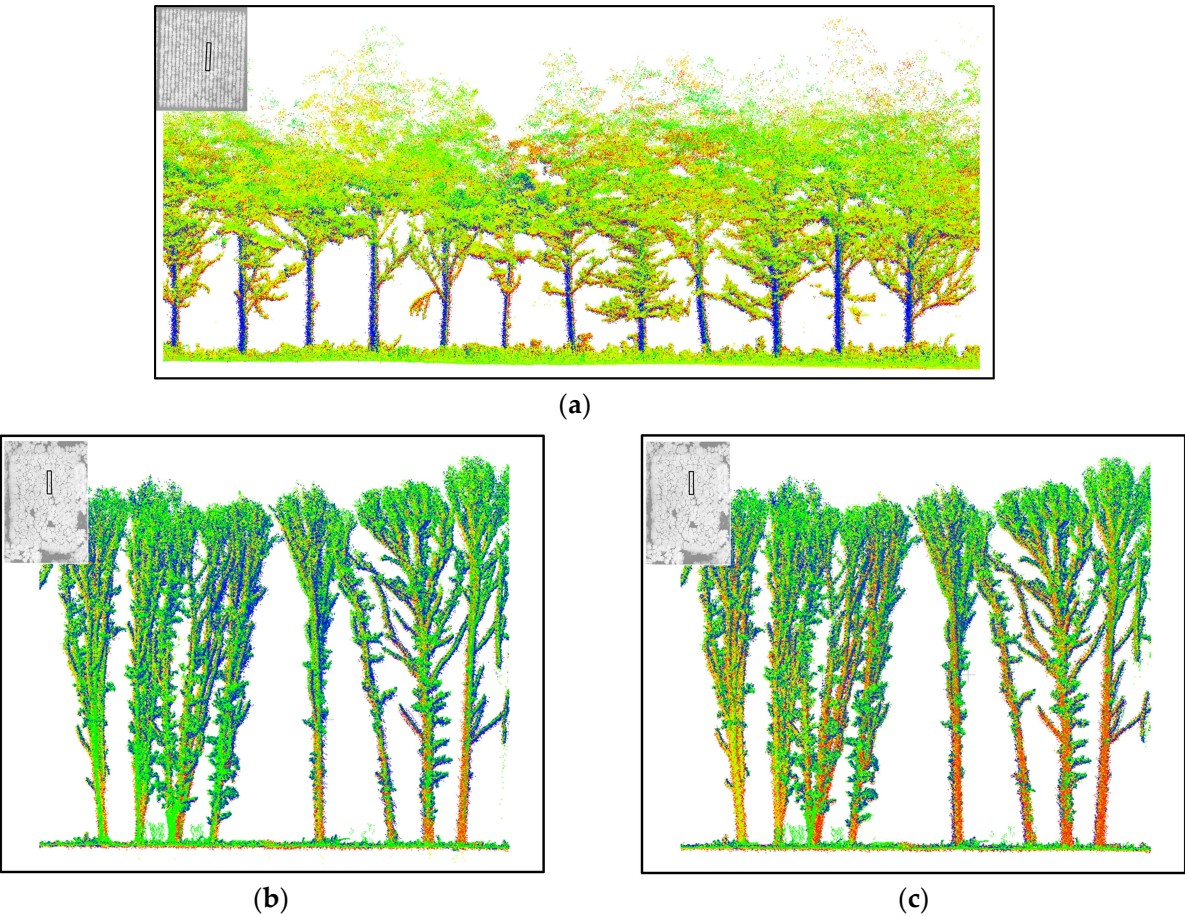

**Figure 16.** Side view of sample profiles (colored by time) after trajectory enhancement depicting the alignment quality for: (**a**) the *YP-BP-2021* dataset, as well as the *MP-BP-2023* dataset from (**b**) Test 1 and (**c**) Test 2.

**Table 5.** Quantitative evaluation of point cloud alignment before and after trajectory enhancement for the Backpack datasets.

| Conducted Test | Point-to-Feature Normal Distance | # Points (Thousands) | Before LSA | | | After LSA | | |
|---|---|---|---|---|---|---|---|---|
| | | | Mean (m) | STD (m) | RMS (m) | Mean (m) | STD (m) | RMS (m) |
| *YP-BP-2021* | Planar Features | 16,789 | 0.224 | 0.171 | 0.282 | 0.026 | 0.021 | 0.034 |
| | Cylindrical Features | 10,805 | 0.190 | 0.181 | 0.262 | 0.016 | 0.017 | 0.024 |
| *MP-BP-2023, Test 1* | Planar Features | 15,002 | 0.530 | 0.804 | 0.963 | 0.026 | 0.032 | 0.041 |
| | Cylindrical Features | 10,329 | 0.472 | 0.388 | 0.611 | 0.029 | 0.041 | 0.050 |
| *MP-BP-2023, Test 2* | Planar Features | 15,002 | 0.530 | 0.804 | 0.963 | 0.034 | 0.044 | 0.055 |
| | Cylindrical Features | 10,329 | 0.472 | 0.388 | 0.611 | 0.034 | 0.043 | 0.055 |

Figure 17 portrays the enhanced trajectory colored by the magnitude of estimated corrections to the position parameters for the Backpack tests, where unadjusted trajectory points (i.e., trajectory epochs that do not correspond to any LiDAR features in the LSA) are colored in grey. Since the entire mission of the *MP-BP-2023* dataset was within the mature plantation, all trajectory points were adjusted. For the *YP-BP-2021* dataset, the figure clearly indicates higher correction magnitudes at the middle portion of the canopy compared with the north and south edges where the GNSS signal reception is better. The largest magnitude of positional corrections is around 1.3 m. In terms of the tests for the *MP-BP-2023* dataset, the corrections were much larger. While no obvious pattern can be observed for Test 1 (Figure 17b), the largest corrections (around 8.6 m) took place at the

middle tracks for Test 2, where drift errors caused by GNSS signal outages accumulated the most (Figure 17c).

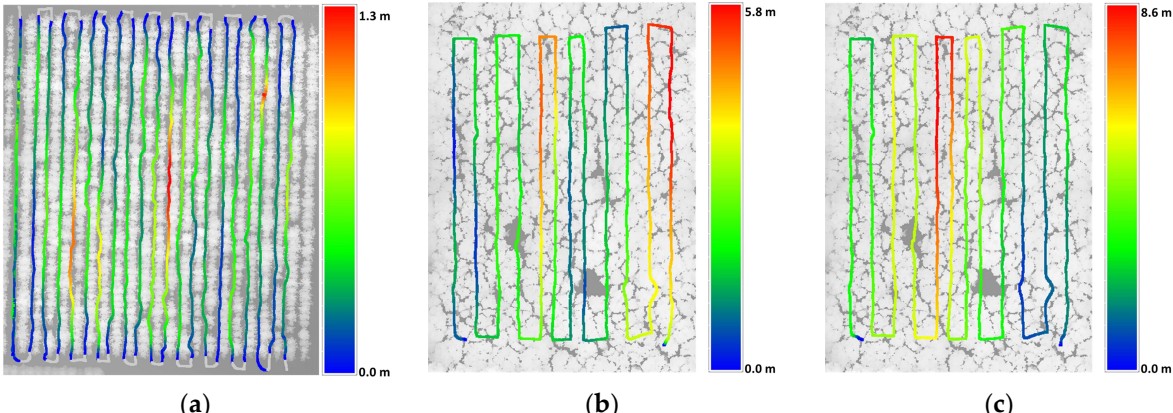

(**a**)            (**b**)            (**c**)

**Figure 17.** Enhanced trajectory for the Backpack datasets colored by the magnitude of interpolated corrections for the position parameters (unadjusted trajectory points are colored in grey) overlaid on the study site's point cloud (colored by height): (**a**) the *YP-BP-2021* dataset, as well as the *MP-BP-2023* dataset from (**b**) Test 1 and (**c**) Test 2.

Statistics of trajectory corrections are presented in Table 6, where the mean and STD values for the adjusted trajectory epochs are reported. The STD values for the position differences of the *YP-BP-2021* dataset were in the range of 20 cm to 30 cm, which were much higher than the values for the UAV datasets (shown in Table 2). For the *MP-BP-2023* dataset, the STD values for the two tests were compatible—within 1 m for the X and Y coordinates and around 2.6 m for the Z direction. However, we can observe a constant correction to the positional components for Test 2 when incorporating reference UAV data in the LSA, while the mean values for Test 1 are close to zero. This indicates a discrepancy between the derived point clouds from Tests 1 and 2 for the *MP-BP-2023* dataset. For the orientation parameters, like the UAV datasets, the heading angle (κ) shows the largest corrections for all experiments. The differences between orientation corrections for these three experiments were not as significant as the ones for the positional components. This signifies that for these datasets, GNSS signal outages affected the positional component of the trajectory more than the orientation one.

**Table 6.** Mean and STD of differences between initial and refined trajectory information for the Backpack datasets.

| | Number of Adjusted Trajectory Epochs | Statistics Measures | $X_{dif}$ (m) | $Y_{dif}$ (m) | $Z_{dif}$ (m) | $\omega_{dif}$ (°) | $\phi_{dif}$ (°) | $\kappa_{dif}$ (°) |
|---|---|---|---|---|---|---|---|---|
| *YP-BP-2021* | 226,500 (100 Hz) | Mean | 0.021 | 0.026 | 0.003 | 0.000 | 0.000 | 0.060 |
| | | STD | 0.186 | 0.279 | 0.261 | 0.005 | 0.026 | 0.115 |
| *MP-BP-2023, Test 1* | 148,500 (100 Hz) | Mean | −0.017 | 0.003 | 0.123 | 0.009 | 0.069 | 0.139 |
| | | STD | 0.976 | 0.554 | 2.567 | 0.060 | 0.059 | 0.191 |
| *MP-BP-2023, Test 2* | 148,500 (100 Hz) | Mean | −0.185 | 0.239 | −3.275 | 0.000 | 0.002 | 0.087 |
| | | STD | 0.985 | 0.578 | 2.603 | 0.033 | 0.034 | 0.191 |

Lastly, the absolute accuracy of the Backpack point clouds after trajectory enhancement was evaluated through a comparison with the refined point cloud from the UAV datasets. Figure 18 shows the sample trees after trajectory enhancement of the Backpack point clouds overlaid with the respective refined UAV point clouds. It can be seen in the figure that the tree trunk from the Backpack point cloud was in good agreement with the UAV data in both X and Y directions for the young plantation datasets, whereas small misalignment can be observed in the Z direction. For the *MP-BP-2023* dataset with poor-quality trajectory,

there was a large shift (around 3.5 m) in the Z direction between the UAV point cloud and refined Backpack point cloud from Test 1. When the reference UAV point cloud was used in the trajectory enhancement process, the derived Backpack point cloud was well-aligned with the UAV data in all directions, as presented in Figure 18c.

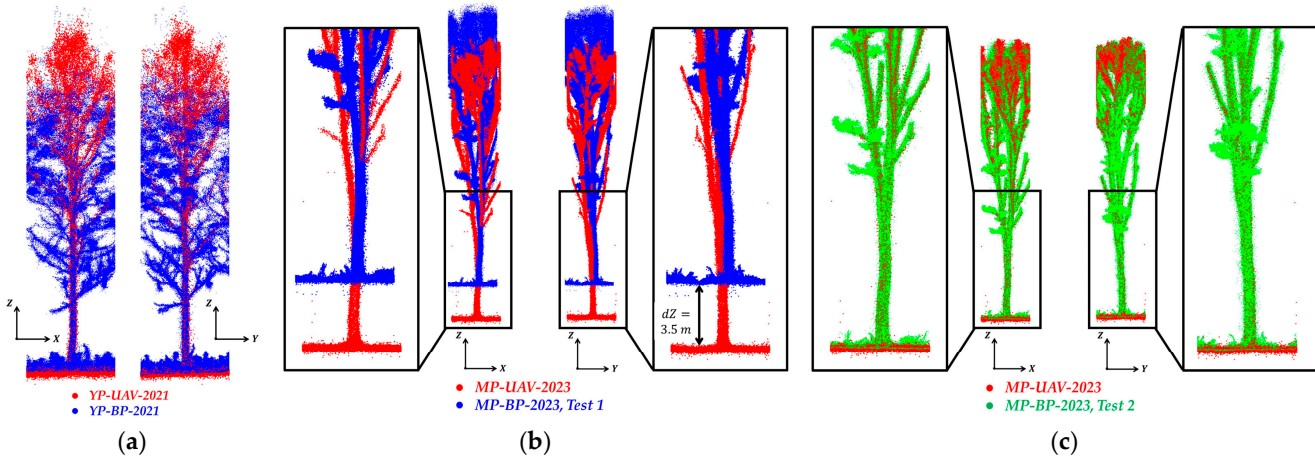

**Figure 18.** A sample tree in the Backpack point clouds after trajectory enhancement overlaid with the refined point cloud from respective UAV datasets (in red) along the X-Z and Y-Z planes: (**a**) the *YP-BP-2021* dataset, as well as the *MP-BP-2023* dataset from (**b**) Test 1 and (**c**) Test 2.

Quantitative comparison between the Backpack and UAV point clouds was carried out using the terrain patches and tree trunks. Table 7 reports the Z differences between the terrain patches as well as X/Y differences and planimetric distances between estimated tree locations from the Backpack and respective UAV datasets. In this table, the alignment in the Z direction suggests that conducted trajectory enhancement of the *YP-BP-2021* dataset achieved a vertical accuracy of 3.4 cm, while the comparison of tree trunks reveals that the tree locations are in agreement with an accuracy of 0.1 m. As for the *MP-BP-2023* dataset with lower-quality trajectories, the misalignment in the horizontal and vertical directions is around 0.2 m and 3.4 m, respectively, for Test 1 using solely Backpack LiDAR data. This misalignment is consistent with the mean corrections to the trajectory positional components for Test 2 (as presented in Table 6). As a result, when incorporating reference UAV point cloud, Test 2 provided results with high absolute accuracy—the alignment in all directions was within 6 cm. In summary, for the Backpack dataset with poor-quality trajectory, a reference point cloud is needed to achieve high absolute accuracy through the proposed trajectory enhancement strategy.

**Table 7.** Quantitative evaluation of the absolute accuracy of the point cloud from the Backpack datasets after trajectory enhancement through a comparison with the reference UAV datasets using extracted terrain patches (for vertical direction) and tree trunks (for planimetric direction).

| Comparison | Statistics Measures | Terrain Patches | Tree Trunks | | |
|---|---|---|---|---|---|
| | | dZ (m) | dX (m) | dY (m) | Planimetric Distance (m) |
| *YP-BP-2021* vs. *YP-UAV-2021* | Mean | 0.004 | −0.005 | −0.028 | 0.078 |
| | STD | 0.033 | 0.061 | 0.070 | 0.059 |
| | RMS | 0.034 | 0.062 | 0.075 | 0.097 |
| *MP-BP-2023 Test 1* vs. *MP-UAV-2023* | Mean | 3.414 | 0.190 | −0.210 | 0.392 |
| | STD | 0.138 | 0.224 | 0.245 | 0.192 |
| | RMS | 3.417 | 0.294 | 0.323 | 0.437 |
| *MP-BP-2023 Test 2* vs. *MP-UAV-2023* | Mean | −0.003 | −0.003 | 0.002 | 0.076 |
| | STD | 0.035 | 0.048 | 0.066 | 0.030 |
| | RMS | 0.035 | 0.048 | 0.066 | 0.082 |

### 5. Conclusions

In this paper, a system-driven framework for system calibration and trajectory enhancement for LiDAR units mounted on UAV and Backpack MMS is proposed to generate accurate point clouds for fine-resolution forest inventory. The strategy starts by reconstructing point clouds using initial system calibration parameters and GNSS/INS trajectory. Terrain patches and tree trunks are then extracted and matched from the LiDAR point clouds. By minimizing the discrepancies among features from different tracks/datasets/systems while considering the absolute and relative positional/rotational information from the initial trajectory, system calibration parameters and trajectory information are refined through a non-linear LSA. This strategy can be conducted on multi-temporal, multi-platform datasets to ensure the best point cloud alignment. Meanwhile, as part of the process, important forest inventory metrics such as tree trunk radius and orientation are derived.

To evaluate the performance of the proposed strategy, three UAV and two Backpack datasets over young and mature plantations were used in this study. For the UAV datasets, sequential system calibration and trajectory enhancement were conducted to improve the accuracy of the point clouds while avoiding any potential correlation among system calibration and trajectory parameters. The results after the system calibration and trajectory enhancement show a reduction in the fitting error for the used terrain patches and tree trunks from 20 cm to 5 cm. The agreement of point clouds from the two UAV datasets over the young plantation reveals that an absolute accuracy in the range of 10 cm was achieved. Overall, it can be concluded that after conducting the LSA using the proposed framework, the point clouds from the UAV systems achieved high levels of relative and absolute accuracy. As for the Backpack datasets, the trajectory of the one collected at the young plantation is of reasonable quality due to its frequent access to open sky. On the other hand, the trajectory related to the dataset at the mature plantation has poor quality. The trajectory enhancement using solely Backpack LiDAR data significantly improved the fitting error of terrain patches and tree trunks in the point cloud from 30 cm to 3 cm and 1 m to 6 cm for the two Backpack datasets. The absolute accuracy of the Backpack point clouds was evaluated through a comparison with the refined UAV point clouds. While using trajectory with reasonable accuracy, the proposed trajectory enhancement approach improved the absolute accuracy of the Backpack point cloud to the 10 cm level. However, when using the lower-quality trajectory, the absolute accuracy of the derived point clouds was poor. After adopting the refined UAV point cloud as a reference, a similar level of absolute accuracy was achieved compared with the dataset with trajectory of reasonable quality.

In summary, the findings of this study for accurate forest stem-level mapping using UAV and Backpack MMS are as follows:

- For applications requiring point clouds within a 20 cm level of accuracy, UAV LiDAR systems with reasonable system calibration parameters and trajectory information can directly provide point clouds that meet such requirements. For applications requiring a 5 cm or better level of accuracy, sequential system calibration and trajectory enhancement is recommended.
- For Backpack systems, the quality of trajectory is affected by GNSS signal outages. Therefore, trajectory enhancement is necessary to improve the quality of point clouds. Frequent access to open sky areas can ensure a reasonable-quality trajectory without a dramatic increase in drifting errors over time. In this case, trajectory enhancement can be conducted without any reference dataset. However, for situations with more severe GNSS signal outages, a reference point cloud (e.g., UAV point cloud) is needed to improve the quality of Backpack point clouds in terms of the absolute accuracy.

The proposed and validated system calibration and trajectory enhancement framework for forest plantations will be used as the foundation for future research targeting accurate under-canopy mapping in rapidly changing natural forest environments. Hence, for future work, the performance of the proposed strategy will be evaluated on natural forests including those with dense canopies, such as tropical areas. In such cases, more severe GNSS signal outages will pose additional challenges for reliable feature extraction and

matching. Modifications to the proposed algorithm will be investigated to increase its robustness to false tree trunk matching in cases where high misalignment within the point cloud might lead to a version of any tree trunk being matched with one of its neighboring trees. Integrating raw IMU measurements, GNSS observations, and RGB imagery with LiDAR will be also explored to provide additional constraints to achieve trajectories with higher accuracy.

**Author Contributions:** Conceptualization, T.Z., R.R., S.F. and A.H.; Data curation, R.M.; Formal analysis, T.Z. and R.R.; Methodology, T.Z., R.R. and Y.-C.L.; Software, T.Z., R.R. and Y.-C.L.; Supervision, S.F. and A.H.; Writing—original draft, T.Z. and R.R.; Writing—review and editing, T.Z., S.F. and A.H. All authors have read and agreed to the published version of the manuscript.

**Funding:** This research was partially supported by the Hardwood Tree Improvement and Regeneration Center, Purdue's Center for Digital Forestry, USDA Forest Service (19-JV-11242305-102), U.S. Department of the Navy, Cooperative Ecosystem Studies Unit program (Agreement # N400852220004), and the U.S. Department of Agriculture, National Institute of Food and Agriculture, Sustainable Agriculture Systems project (2023-68012-38992).

**Data Availability Statement:** Data sharing is not applicable to this paper.

**Acknowledgments:** The authors would like to thank the Academic Editor and four anonymous reviewers for providing helpful comments and suggestions which substantially improved the manuscript.

**Conflicts of Interest:** The authors declare no conflict of interest.

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
