# Peer review of "In Situ Calibration and Trajectory Enhancement of UAV and Backpack LiDAR Systems for Fine-Resolution Forest Inventory"

_remotesensing, doi:10.3390/rs15112799_

Round 1

Reviewer 1 Report

My congratulations for the work done. The work is well organized and easy to read. The inputs are well established and demonstrated.

I would like the data to be public, so future researches can use it.

Author Response

Thank you for your valuable comments, please find the response attached.

Reviewer 2 Report

The manuscript presents new method for increasing relative and absolute accuracy of LiDAR data collected by UAV and backpack based devices.

The methods for sequential calibration and trajectory enhancement are well defined and presented with very high level of detail. 

The results are also presented in detail and show promising accuracy improvements for the usage in forest inventory.

There are some misalignments in display of tables and figures, that should be addressed in the final version of manuscript. 

Quality of English language is good.

Author Response

(The authors gave the same response as above.)

Reviewer 3 Report

Previous work proposes forest remote sensing inventories based on 3D point clouds combining UAV-LiDAR (above the forest) and backpacks with LiDAR units, GNSS geolocation, and Inertial Measurement Unit (IMU) providing cloud points below the canopy and fine-resolution forest metrics such as the diameter at breast height (DBH), tree height (H), canopy cover (CC), etc. However, GNSS outages (no clear skies) and LiDAR mounting inaccuracies lead to misalignments within the cloud points. A new framework for system calibration and trajectory enhancing of the UAV and backpack trajectories is proposed.

The introduction in clear, the methodology is appropriated. Validation is performed for the UAVs and backpacks by the alignment error of two cloud points in the young and mature plantations.

The Tables are clear and of good quality. The figures are of excellent quality, fine details can be seen on them. Results are well presented.

Results are well presented and clearly show a very good improvement reducing errors after trajectory enhancement for both UAVs and backpacks.

Conclusions summarize well the results. However, the forest  corresponds to a plantation (stand) with enough separation along each axis, although the mature plantation canopy overlaps the dataset was capture under leaf-off conditions, so the sky is visible. What would happen on natural dense forests such as the tropic where leaf-off conditions are not possible? Please analyze these cases in the conclusions.

Author Response

(The authors gave the same response as above.)

Reviewer 4 Report

IMPORTANT: Some sections (few sentences) show high similarity in Turnitin check (file attached). Authors may rewrite some of the highlighted text to reduce similarity. 

Page-2, Page-4, Page-5, Page-6, Page-12, Page-13

Overall, the manuscript is well-written and acknowledges its limitations. The synergistic properties of unmanned aerial vehicle and mobile ground mapping systems have been investigated. A system-driven strategy for mounting parameters estimation and trajectory enhancement using terrain regions and tree trunks is proposed to improve the quality of acquired point clouds. In addition, forest inventory metrics such as tree trunk radius and orientation are derived. The proposed strategy reduces misalignment within point clouds resulting from inaccurate LiDAR mounting parameters and/or GNSS signal disruptions. 

One of the key issue between 2001 and 2002 dataset have been handled reasonably. The lower geometric accuracy for the YP-UAV- 2022 dataset is expected, as described by the authors.

In forest environments, there are common features that can be autonomously identified and extracted from point clouds. To refine system calibration and trajectory parameters, terrain regions, and tree trunks are extracted and utilized as planar and cylindrical features, respectively. Individual tree detection and localization followed by tree trunk extraction and matching have been achieved with reasonable accuracy. One of the key contributions of the proposed strategy is the provision of vital forest inventory biometrics, such as tree trunk radius and orientation. 

The known differences (tree thinning activity) between the YP-UAV-2021 and YP-UAV-2022 datasets have been accounted for in a manner that ensures accurate tree accounting.

The well-aligned point clouds from the two UAV datasets over the youth plantation indicate that the system calibration and trajectory enhancement framework achieved high absolute accuracy. The locations of trees coincide with an accuracy of 0.1 m

The following sections are blank.

Funding: 

Data Availability Statement: 

Acknowledgments: 

Conflicts of Interest:

Author Response

(The authors gave the same response as above.)
